# WNK1–OSR1 Signaling Regulates Angiogenesis-Mediated Metastasis towards Developing a Combinatorial Anti-Cancer Strategy

**DOI:** 10.3390/ijms232012100

**Published:** 2022-10-11

**Authors:** Chia-Ying Hou, Chung-Yung Ma, Yu-Ju Lin, Chou-Long Huang, Horng-Dar Wang, Chiou-Hwa Yuh

**Affiliations:** 1Institute of Molecular and Genomic Medicine, National Health Research Institutes, Zhunan, Miaoli County 35053, Taiwan; 2Institute of Biotechnology, National Tsing Hua University, Hsinchu 300044, Taiwan; 3Division of Nephrology, Department of Internal Medicine, University of Iowa Carver College of Medicine, Iowa City, IA 52242, USA; 4Institute of Systems Neuroscience, National Tsing Hua University, Hsinchu 300044, Taiwan; 5Department of Life Science, National Tsing Hua University, Hsinchu 300044, Taiwan; 6Institute of Bioinformatics and Structural Biology, National Tsing Hua University, Hsinchu 300044, Taiwan; 7Department of Biological Science and Technology, National Yang Ming Chiao Tung University, Hsinchu 30010, Taiwan; 8Ph.D. Program in Environmental and Occupational Medicine, Kaohsiung Medical University, Kaohsiung 80708, Taiwan

**Keywords:** lysine-deficient protein kinase-1 (WNK1), oxidative stress responsive 1 (OSR1), Protein Phosphatase 2A (PP2A), tumor-induced angiogenesis, hepatocellular carcinoma (HCC)

## Abstract

Lysine-deficient protein kinase-1 (WNK1) is critical for both embryonic angiogenesis and tumor-induced angiogenesis. However, the downstream effectors of WNK1 during these processes remain ambiguous. In this study, we identified that oxidative stress responsive 1b (*osr1b*) is upregulated in endothelial cells in both embryonic and tumor-induced angiogenesis in zebrafish, accompanied by downregulation of protein phosphatase 2A (pp2a) subunit *ppp2r1bb*. In addition, *wnk1a* and *osr1b* are upregulated in two liver cancer transgenic fish models: [*tert x p53^−/−^*] and [*HBx,src,p53^−/−^,RPIA*], while *ppp2r1bb* is downregulated in [*tert x p53^−/−^*]. Furthermore, using HUVEC endothelial cells co-cultured with HepG2 hepatoma cells, we confirmed that WNK1 plays a critical role in the induction of hepatoma cell migration in both endothelial cells and hepatoma cells. Moreover, overexpression of OSR1 can rescue the reduced cell migration caused by shWNK1 knockdown in HUVEC cells, indicating OSR1 is downstream of WNK1 in endothelial cells promoting hepatoma cell migration. Overexpression of PPP2R1A can rescue the increased cell migration caused by WNK1 overexpression in HepG2, indicating that PPP2R1A is a downstream effector in hepatoma. The combinatorial treatment with WNK1 inhibitor (WNK463) and OSR1 inhibitor (Rafoxanide) plus oligo-fucoidan via oral gavage to feed [*HBx,src,p53^−/−^,RPIA*] transgenic fish exhibits much more significant anticancer efficacy than Regorafenib for advanced HCC. Importantly, oligo-fucoidan can reduce the cell senescence marker-IL-1β expression. Furthermore, oligo-fucoidan reduces the increased cell senescence-associated β-galactosidase activity in *tert* transgenic fish treated with WNK1-OSR1 inhibitors. Our results reveal the WNK1–OSR1–PPP2R1A axis plays a critical role in both endothelial and hepatoma cells during tumor-induced angiogenesis promoting cancer cell migration. By in vitro and in vivo experiments, we further uncover the molecular mechanisms of WNK1 and its downstream effectors during tumor-induced angiogenesis. Targeting WNK1–OSR1-mediated anti-angiogenesis and anti-cancer activity, the undesired inflammation response caused by inhibiting WNK1–OSR1 can be attenuated by the combination therapy with oligo-fucoidan and may improve the efficacy.

## 1. Introduction

Cancer is one of the leading causes of death globally and its incidence is still rising [1]. Liver cancer ranks fourth among cancers in terms of mortality and incidence [2], and there is a lack of effective systematic treatments for advanced-stage of liver cancer [3]. Effective treatment for liver cancer is an urgent unmet medical need. Liver cancer is a highly-vascularized tumor with strong angiogenetic activity [4], and as such, targeting tumor-induced angiogenesis has been suggested as a potential anti-cancer strategy. In comparison to cancer that exhibits heterogeneity and genomic instability, the DNA of endothelial cells is not easily mutated, suggesting that it is unlikely to produce drug resistance in long-term drug usage against angiogenesis [5]. Furthermore, considering its lower toxicity and fewer side effects in tumor treatments, target therapy against tumor-induced angiogenesis is a promising means to cure cancer.

The “with-no-lysine kinase” (WNK) is a subfamily of serine/threonine kinase with the atypical placement of the catalytic lysine in subdomain II (Cys^250^) instead of subdomain I [6]. WNK1 has three homologs in humans—WNK2, WNK3, and WNK4, and it is widely expressed in many organs including the kidney, heart, liver, and spleen [7]. WNK1 kinase can bind to the conserved CCT domains of SPS/Ste20-related proline–alanine-rich kinase (SPAK) and oxidative stress-responsive kinase 1 (OSR1) through its RFXV (arginine-phenylalanine-any amino acid-valine) motif and phosphorylates them [8]. OSR1 and SPAK can phosphorylate and regulate many ion transporters and cation-chloride co-transporter (CCC) including Na+-K+-2Cl− cotransporter (NKCC), K+-Cl− cotransporter (KCC), Na+-Cl− cotransporter (NCC), and renal outer medullary potassium channel (ROMK). Consequently, WNK1-OSR1/SPAK pathways regulate renal epithelial ion transport, cell volume, and ion homeostasis [9]. WNK1 plays an important role in ion homeostasis and mutation/overexpression of WNK1 leads to serious diseases as consequence.

WNK1 is involved in many cancers including liver cancer, breast cancer, glioblastoma, giant cell tumors, retinoblastoma, clear-cell renal-cell carcinoma, and lung adenocarcinoma [10,11,12,13,14,15,16,17]. NKCC1 overexpression enhances the proliferation and invasion of HCC cells through the WNK1–OSR1–NKCC1 pathway [18]. In HCC patients, higher WNK1 expression is associated with poor prognosis [11]. In breast cancer, WNK1 regulates AXL, a tyrosine kinase related to metastasis, independent of OSR1 [12]. SPAK is a downstream effector of WNK1 enhancing EMT. Through interacting with SPAK, SNAIL is stabilized and increases nuclear retention [19]. OSR1 interacts with SMAD2/3 and promotes its nuclear translocation, enhancing TGF-β1 signaling and facilitating EMT [13].

In addition to activating signaling pathways in cancer cells, WNK1 might participate in cancer formation through angiogenesis. WNK1 was detected with strong expression in the early development of the cardiovascular system and in endothelial cells, the vascular smooth muscle cells of adult vessels [20]. Wnk1 homozygous mutant mice died before day 13 of gestation with aberrant embryonic angiogenesis and cardiac defects, which can be rescued by endothelial expression of Wnk1, indicating that Wnk1 is involved in the late stage of angiogenesis [21]. Furthermore, the angiogenesis and cardiac defects in Wnk1 null embryos can be rescued by the expression of the constitutive active Osr1 in endothelial cells [22]. Wnk1 deletion in the uterine of mice caused vascularization deficiency on gestation day 8.5. This is one of the reasons contributing the abnormal implantation and compromised fertility [23]. These results conclude that the WNK1–OSR1 axis is involved in embryonic angiogenesis and development. In zebrafish, wnk1a and wnk1b are two paralogues of human WNK1. Both of them are involved in embryonic angiogenesis through the VEGFR–PI3K–AKT pathway [24]. However, whether WNK1 affects embryonic angiogenesis through its downstream OSR1, or other effectors, remains unknown. Thus, it prompts us to explore this issue.

WNK1 is involved in cell migration partly through its ability in ion regulation. During cellular migration, the volume of cells changes, which mainly depends on the activities of the ion transporter and cotransporters. By regulating cellular [Na^+^], [K^+^], [Cl^−^], water uptake on the front end of the cell, and release at the rear end. The circulation of water uptake and release forms a force permitting cells to move [25]. WNK1 is also involved in many pathways, including MAPK, WNT/β-catenin, TGF-β-SMAD2, 5′AMP-activated protein kinase, and PP2A subunit alpha isoform, which contribute to cell proliferation, migration, and angiogenesis, and thus is associated with cancer progression [26]. PPP2R1A, a subunit of Ser/Thr protein phosphatase 2A (PP2A), has been found to act as a tumor suppressor. Mutations of PPP2R1A occur in several cancers to promote malignant cell growth [27]. Nonetheless, whether PPP2R1A is downstream of WNK1 in angiogenesis and cancer remains obscure.

WNK1 and its downstream effectors are considered to be potential diagnostic and therapeutic targets. In HCC, elevated WNK1 expression showed poorer overall survival (OS) [11]. In breast cancer, high pOSR1 levels manifest poor OS and distant disease-free survival (DDFS) [13]. WNK463 inhibits the kinase activity in the WNK family (WNK1, WNK2, WNK3, WNK4). Through binding to WNK1 and WNK4, WNK463 affects electrolyte excretion and decreases blood pressure effectively [28]. Rafoxanide and Closantel are inhibitors of OSR1/SPAK; they both target the CCT domains of OSR1/SPAK, preventing WNK1 binding and activation [29]. WNK463 and Closantel reduced the HCC formation in the zebrafish HCC model driven by [*HBx, Src, p53^−^*] crossed with endothelial WNK1 overexpression [15]. In this study, we further explored the function of the WNK1–OSR1 axis both in hepatoma cells and endothelial cells using in vitro co-culture HUVEC and hepatoma cells. Previous reports have indicated that loss of WNK appears to be pro-inflammatory, and thus targeting WNK1 might induce inflammation [30]. Oligo-fucoidan, sulfated polysaccharides extracted from brown seaweed, exhibits an anti-cancer effect [31]. Oligo-fucoidan is also known for its immune modulation and anti-inflammatory function. Our previous study indicated that oligo-fucoidan possesses a hepatoprotective effect [32]. In this study, we treated [*HBx, Src, RPIA, p53^−/−^*] transgenic zebrafish which develop HCC at 5 months of age, with WNK1–SPAK/OSR1 axis inhibitors combined with oligo-fucoidan to further improve the efficacy of WNK1–OSR1 axis inhibitors combined with oligo-fucoidan to protect the normal cells. We wish to develop a novel combination with a more effective therapeutic effect against high-mortality HCC.

In this study, we used xenotransplantation to investigate the involvement of WNK1–OSR1–PPP2R1A during tumor-induced angiogenesis, via [*tert x p53^−/−^*] to observe the markers of angiogenesis and cell proliferation during carcinogenesis, and through feeding [*HBx,src,p53^−/−^,RPIA*] transgenic fish with drugs to examine the WNK–OSR1 inhibitors combination therapy with oligo-fucoidan. Our results reveal the WNK1–OSR1–PPP2R1A axis plays a critical role in both endothelial and hepatoma cells during tumor-induced angiogenesis promoting cancer cell migration, and uncover the molecular mechanisms with WNK1 and its downstream effectors during tumor-induced angiogenesis. Importantly, we provide a more effective combinatorial treatment as a potential anticancer strategy.

## 2. Results

### 2.1. The Expression of wnk1a–osr1a/osr1b Is Upregulated during Embryonic Angiogenesis

WNK1 is known to play a critical role in embryonic angiogenesis [24]. To find out the downstream effectors of WNK1, we first examined the expression of WNK1 and the possible downstream effectors during embryonic angiogenesis in zebrafish. As embryonic angiogenesis occurs between 24 and 48 h, the 24, 36, 48 h post fertility (hpf) embryos were collected, and the expression of the WNK1–OSR1 axis was examined by qPCR. We found that *wnk1a*, a paralog of *WNK1*, was upregulated at 36 and 48 hpf (Figure 1a). The expressions of *osr1a* and *osr1b*, paralogs of *OSR1*, were upregulated at 36 hpf, but the expression declined at 48 hpf (Figure 1b,c). However, the expression of *stk39*, a paralog of *SPAK1*, was downregulated at 36 and 48 hpf (Figure 1d). Because WNK1 activates OSR1/SPAK, if the *osr1a*/*osr1b* or *stk39* are involved in embryonic angiogenesis, they are supposed to be upregulated; thus, the results indicated that *wnk1a*-*osr1a*/*osr1b* might be involved in embryonic angiogenesis, but not *stk39*. We also measured the expression of *ppp2r1ba* and *ppp2r1bb*, two paralogs of *PPP2R1A* in zebrafish embryos. PPP2R1A, a subunit of PP2A which is a serine/threonine phosphatase, is the candidate to repress the WNK1–OSR1 axis. The expressions of *ppp2r1ba* and *ppp2r1bb* were upregulated at 36 hpf (Figure 1e,f). Because WNK1 can be inactivated by PPP2R1A, if *ppp2r1ba* and *ppp2r1bb* are involved in embryonic angiogenesis, we expected they should be down-regulated, so those results indicated that *ppp2r1ba* and *ppp2r1bb* might not be involved in embryonic angiogenesis.

Those data above suggest that the downstream effectors of *WNK1*: *osr1a* and *osr1b* might play important roles in embryonic angiogenesis since they expressed between 24 to 48 hpf, but *stk39*, *ppp2r1ba* and *ppp2r1bb* might not be involved in embryonic angiogenesis.

### 2.2. The Expression of wnk1a and osr1b Is Upregulated in Endothelial Cells, and WNK1 and OSR1 Are Upregulated in Hepatoma Cells during Tumor-Induced Angiogenesis

The WNK1–OSR1 axis might have different roles in embryonic angiogenesis versus tumor-induced angiogenesis. To further understand which downstream targets of WNK1 play critical roles during tumor-induced angiogenesis, we performed xenotransplantation and measured the expression of genes in endothelial cells and hepatoma cells. By injecting human hepatoma cell Hep3B into zebrafish embryos, we could easily distinguish the gene expression profile between hepatoma cells and endothelial cells by designing primers specific to the human and zebrafish genes for QPCR.

Firstly, we must make sure the angiogenesis stimulates hepatoma cell proliferation in the xenotransplantation model. We observed that the green fluorescence endothelial cells grow toward hepatoma cells from 1 day post injection (dpi) to 3 dpi, and the red fluorescence hepatoma cells increased in area from 1 dpi to 3 dpi (Figure 2a). We also injected the Hep3B cells into Tg(fli1:wnk1a;myl7:EGFP) fish (WNK1(OE)), and found the cell proliferation was dramatically increased in 3 dpi compared to the control Tg(fli1:EGFP) fish. Furthermore, Hep3B cells were microinjected into Tg (fli1:CreERT2;myl7:EGFP)xTg(loxP-wnk1a-DsRed-loxP) embryos after RU486 treatment to induce CreER-mediated wnk1a knockout. We found the blood vesicles were much fewer in WNK1(KO) fish, and the Hep3B cell proliferation was significantly reduced compared to the control and WNK1 (OE) fish (Figure 2a). These results indicated that *wnk1a* in endothelial cells plays an important role in hepatoma cell proliferation. We also compared the migration behavior of Hep3B cells in WNK1 overexpression or knockdown in endothelial cells, and the results showed WNK1 overexpression in endothelial cells promoting hepatoma cell migration, while WNK1 knockdown in endothelial cells reduced the hepatoma cell migration (Figure 2b). Our data suggest that WNK1 in endothelial cells enhances hepatoma cell proliferation and migration.

Next, we measured the expression of WNK1 and its downstream effectors in endothelial cells using zebrafish-specific primers. Our qPCR analysis indicated that *wnk1a* was upregulated from 1 dpi to 2 dpi (Figure 3a), indicating *wnk1a* is an immediate early gene required for tumor-induced angiogenesis. Interestingly, *osr1a* was downregulated from 2 dpi to 3dpi (Figure 3b), but *osr1b* was consistently upregulated from 1 to 3 dpi, and the folds of increment were dramatic (Figure 3c). The expression of *stk39* was upregulated in the beginning at 1 dpi, but then downregulated at 2 dpi and 3 dpi (Figure 3d). Because WNK1 activates OSR1/SPAK, if the *osr1a*/*osr1b* or *stk39* are involved in tumor-induced angiogenesis, they are supposed to be upregulated from 2 dpi to 3 dpi. Since we used zebrafish-specific primers, these results suggest that *wnk1a* and *osr1b* in endothelial cells are involved in tumor-induced angiogenesis. Moreover, *ppp2r1ba* and *ppp2r1bb* were first upregulated at 1 dpi, but were dramatically reduced at 3 dpi (Figure 3e,f). Because WNK1 can be inactivated by PPP2R1A, the diminished expressions of *ppp2r1ba* and *ppp2r1bb* at 3 dpi might be pathogenic in the time ahead, so these results suggest the vital roles of *wnk1a*, *osr1b*, *ppp2r1ba*, and *ppp2r1bb* in endothelial cells might be involved in tumor-induced angiogenesis, which stimulates hepatoma cell proliferation.

Subsequently, we detected the expression of WNK1 and its downstream effectors in hepatoma cells using human-specific primers. From QPCR analysis, we found *WNK1* was significantly upregulated at 1 dpi and 3 dpi (Figure 4a), and *OSR1* was dramatically upregulated at 3 dpi for more than 20 folds (Figure 4b). However, *STK39* was first downregulated, then became upregulated at 3 dpi for 2 folds (Figure 4c). Because WNK1 activates OSR1/SPAK, if OSR1 or STK39 are involved in tumor-induced angiogenesis, they are supposed to be upregulated from 2 dpi to 3 dpi, so these results indicate that WNK1 and OSR1 in hepatoma cells are involved in tumor-induced angiogenesis. Nonetheless, there is no difference in the expression of *PPP2R1B* (Figure 4d). Those results imply the importance of WNK1 and OSR1 in hepatoma cell during angiogenesis stimulated hepatoma cell proliferation, but not *PPP2R1B* and *STK39*.

Our results support the working model that the WNK1–SPAK/OSR1 axis is indeed involved in tumor-induced angiogenesis in both cancer cells and endothelial cells. The expression of *wnk1a* and *osr1b* is upregulated in endothelial cells, and *WNK1* and *OSR1* are upregulated in hepatoma cells during tumor-induced angiogenesis.

### 2.3. [tert] and [tert x p53-/-] Transgenic Fish Develops HCC at 15 and 30 dpf

Xenotransplantation might result in different liver cancer formation to that which occurs in reality, and WNK1 has been found to act as an oncogene in many cancers. In order to understand which downstream targets of WNK1 play critical roles during hepatocarcinogenesis, we established [*tert*] and [*tert x p53^−/−^*] fish. Our previous study demonstrated that liver-specific *tert* overexpression can lead to HCC in 15 dpf. Sustaining chronic proliferation is abnormal and a hallmark of cancer [33]. Cyclin E1 (CCNE1) plays an important role in the G1/S phase transition of the cell cycle and has been found to play an oncogenic role in many cancers, such as triple-negative breast cancer, epithelial ovarian cancer, and liver cancer [34,35,36]. Cyclin-dependent kinase 1 (Cdk1) is a contributor to driving G2/M phase transition and has proven essential for cell proliferation and tumorigenesis in liver cancer [37]. CCNE1 and CDK2 are critical for the initiation of HCC [38].

We confirmed that 15 dpf [*tert*] transgenic fish exhibit increased expression of proliferation markers—*ccne1*, *cdk1*, and *cdk2* by QPCR (Figure 5a–c). However, the tumorigenesis of [*tert*] transgenic fish vanishes at 30 dpf (Figure 5a–c). Simultaneously, we observed upregulation of p53. P53 is a tumor suppressor, and previous studies indicate that hTERT transcription was downregulated by p53, which inhibited telomerase activity by repressing the SP1 binding to the TERT promoter [39]. We attempt to solve this problem by crossing the *tert* overexpression fish with *p53* mutant fish to get [*tert x p53^−/−^*] transgenic fish. We found the expression of *ccne1*, and *cdk2* was significantly higher in [*tert x p53^−/−^*] transgenic fish than WT at 30 dpf, when the expression of cell proliferation markers was undetectable in [*tert*] transgenic fish (Figure 5d,f). This result suggested that *tert* collaborating with p53^−/−^ mutation promotes cell proliferation in hepatocytes in 30 dpf. Histopathological examination by H&E staining was used to further verify this finding. The malignant cell characterized by hypertrophy, mitoses, and atypical mitosis forms [40]. The ratios of mitotic figure, trinucleated, and karyomegalic cells of [*tert x p53^−/−^*] transgenic fish were remarkably higher than the hepatocyte from wild-type zebrafish (Figure 5g–j). We conclude that [*tert x p53^−/−^*] transgenic fish can maintain tumorigenesis at 30 dpf according to the QPCR and H&E staining.

### 2.4. Expression of wnk1a, osr1b and stk39 Is Increased, and ppp2r1ba and ppp2r1bb Are Decreased in HCC Formation in the [tert] and [tert x p53^−/−^] Transgenic Fish

To further understand which downstream targets of WNK1 play critical roles during hepatocarcinogenesis, we measured the expression of *wnk1a* and its downstream effectors in [*tert*] and [*tert x p53^−/−^*] fish HCC model. We found the expression of *wnk1a* was notably higher in both [*tert*] and [*tert x p53^−/−^*] transgenic fish than in WT (Figure 6a). Due to inducing angiogenesis being a hallmark of cancer [33], upregulated *wnk1a* expression further implied [*tert*] and [*tert x p53^−/−^*] transgenic fish were undergoing tumorigenesis in 15 dpf and that the WNK1 axis indeed participates in tumor-induced angiogenesis. Although the expression of *osr1a* was significantly downregulated (Figure 6b), the expression of *osr1b* was dramatically increased in both [*tert*] and [*tert x p53^−/−^*] fish (Figure 6c). Distinct from xenotransplantation models, the expression of *stk39* was significantly higher in [*tert*] and [*tert x p53^−/−^*] transgenic fish (Figure 6d). As for *pppsr1ba* and *ppp2r1bb*, their expression declined in both [*tert*] and [*tert x p53^−/−^*] (Figure 6e,f).

Our results support the working model that the WNK1–SPAK/OSR1 axis is indeed involved in liver cancer formation in the [*tert*] and [*tert x p53^−/−^*] transgenic fish models. Obviously, the expression of *wnk1a*, *osr1b,* and *stk39* is upregulated, while *pppsr1ba* and *ppp2r1bb* are downregulated in zebrafish when hepatocytes increase proliferation and display histopathological changes as hepatocellular carcinoma.

### 2.5. Expression of wnk1a and osr1b Is Increased in HCC Formation in the [HBx,src,p53^−/−^,RPIA] Transgenic Fish

As [*tert*] and [*tert x p53^−/−^*] fish might present a distinct subtype of HCC, to advance our understanding of the WNK1 downstream effectors during HCC formation, we utilized [*HBx,src,p53^−/−^,RPIA*] transgenic fish, which is considered as an adult HCC fish model. Previously we have established [*HBx,src,p53^−/−^,RPIA*] transgenic fish for the HCC model and found it develops HCC at 5 months of age. Correspondingly, we observed mitotic figure in [*HBx,src,p53^−/−^,RPIA*] transgenic fish from 1 month of age. Trinucleated cells appeared from 3 months of age and karyomegalic cells were discovered at 5 months of age (Figure 7a). The expression of proliferation markers—*ccne1*, *cdk1*, and *cdk2*—was significantly higher than WT at 5M (Figure 7b).

We measured the expression of the WNK1–SPAK/OSR1 axis and PP2A subunit A in [*HBx,src,p53^−/−^,RPIA*] transgenic fish, and found the expression of *wnk1a* and its downstream effectors—*osr1a*, *osr1b* and *stk39*—was increased compared with WT (Figure 7c–f). Different from the expression pattern of [*tert*] and [*tert x p53^−/−^*] at 15dpf, the expression of *ppp2r1ba* and *ppp2r1bb* was upregulated (Figure 7g,h); this unexpected result implied that there are other targets for *ppp2r1ba* and *ppp2r1bb* in this independent HCC model.

Our results support the working model that the WNK1–SPAK/OSR1 axis is indeed involved in liver cancer formation in the [*HBx,src,p53^−/−^,RPIA*] transgenic fish model. Certainly, the expression of *wnk1a*, *stk39*, *osr1a*, and *osr1b* is upregulated when hepatocytes increase proliferation and display histopathological changes as hepatocellular carcinoma. However, *pppsr1ba* and *ppp2r1bb* seem to play less critical roles in this adult HCC model compared to the [*tert*] and [*tert x p53^−/−^*] transgenic larvae HCC models.

### 2.6. Endothelial Cells Promote Hepatoma Cell Migration via WNK1–OSR1 axis

For the mechanistic study of the WNK1 axis in endothelial cells and cancer cells, we co-cultured the human umbilical vein endothelial cells (HUVEC) with HepG2 hepatoma cells and performed a transwell migration assay. We modulated the expression of WNK1 and OSR1 in HUVEC. To determine the effective knockdown of WNK1 through transfection and the impact on downstream mediators, we performed Western blot analysis for HUVEC cells by WNK1 knockdown, OSR1 overexpression, and together (Figure 8a–c). HUVEC cells were seeded in the lower chamber while HepG2 were seeded in the upper chamber for the transwell migration assay (Figure 8d). After 18 h, crystal violet staining was performed to observe the migrated hepatoma cells, and the representative images are shown in Figure 8e. After quantification, we found that the presence of HUVEC cells stimulates hepatoma cell migration compared to no HUVEC cells. Knockdown of WNK1 in HUVEC cells with two independent shRNAs targeting on WNK1 (shWNK1-1 and shWNK1-2) significantly decreased the migrated hepatoma cells (Figure 8f), WNK1 overexpression in HUVEC cells significantly increased the HepG2 migration (Figure 9). These results indicate that endothelial WNK1 plays an important role in stimulating hepatoma cell migration. In addition, OSR1 overexpression in HUVEC cells can rescue the reduction of migrated cells caused by shWNK1, indicating that OSR1 is a downstream effector of endothelial WNK1 promoting hepatoma cell migration (Figure 8f). We noticed that OSR1 overexpression showed no difference in migration stimulation, perhaps because OSR1 is already abundant in the HUVEC cells. We also found that the knockdown of WNK1 decreased OSR1 protein levels (Figure 8a–c), yet the underlying mechanism is still unclear. Nevertheless, these data suggest that endothelial cells promote hepatoma cell migration via the WNK1–OSR1 axis.

### 2.7. PPP2R1A Acts as a Repressor of WNK1 in Stimulating Hepatoma Cell Migration

PPP2R1A, the subunit of PP2A, has been reported to dephosphorylate and inactivate WNK1, while mutant PPP2R1A increases pWNK in gastrointestinal stromal tumors [41]. To further address the role of PPP2R1A in WNK1 in hepatoma cell migration, we transfected HepG2 cells with WNK1 and PPP2R1A, and co-cultured them with the HUVEC cells for a transwell migration assay. To determine the effective overexpression of WNK1 and PPP2R1A through transfection and the impact on downstream mediators, we performed Western blot analysis for HepG2 cells overexpressing WNK1, PPP2R1A, and together (Figure 10a–c). The HUVEC cells were seeded in the lower chamber and the transfected hepatoma cells (HepG2) were seeded in the upper chamber (Figure 10d). After 18 h, crystal violet staining was performed to observe the migrated hepatoma cells, and the representative images and quantification are shown (Figure 10e,f). Overexpressing WNK1 in HepG2 cells significantly enhanced the migrated cells, indicating that WNK1 is noteworthy in hepatoma cells for stimulating hepatoma cell migration. Interestingly, simultaneous overexpression of PPP2R1A in HepG2 cells can block the enhanced migrated cells upon WNK1 overexpression, indicating that PPP2R1A may have the ability to dephosphorylate and inactivate the WNK1–OSR1 axis (Figure 10f). Overexpression of PPP2R1A in HepG2 also significantly increased the migrated cells, which was unexpected. It could be there are other targets for PPP2R1A in hepatoma cells, some of which might be oncogenic.

The results suggest that PPP2R1A acts as a repressor of WNK1 in stimulating the migration of hepatoma cells. Whether PPP2R1A is targeting WNK1, OSR1, or both, it requires Western blot to detect the phosphorylated protein. Nonetheless, PP2A can target pERK, and reduced PP2A promotes and activates the ERK signal pathway in cancer formation. Increased PP2A might inhibit tumor migration and proliferation.

### 2.8. Combinational Therapy Targeting WNK1–OSR1 with Oligo-Fucoidan Attenuates HCC Formation in [HBx,src,p53^−/−^,RPIA] Transgenic Fish

Several studies have revealed that the WNK1–OSR1/SPAK axis is considered as a potential target for anti-cancer therapy [13,15]. Our previous study also showed WNK1-OSR1 inhibitors, WNK463 and Closantel, reduced [*HBx,src,p53^−/−^*]-DIO HCC and [RPIA] CRC formation in zebrafish models [15]. However, the high toxicity of Closantel is a potential problem for therapy. Furthermore, it was reported that the inhibition of WNK1–OSR1 might cause inflammation [30]. To solve this problem, we tested the combination therapy with WNK1/OSR1 inhibitors and oligo-fucoidan. Oligo-fucoidan was found to have anti-HCC [32] and hepato-protection effects [42], as well as an anti-angiogenesis function [43], and thus it may have a synergistic effect of enhancing anti-HCC efficacy by combining with WNK1/OSR1 inhibitors.

We used [*HBx,src,p53^−/−^,RPIA*] transgenic fish which develops HCC at 5 month of age as an HCC model. We divided the fish into four groups and used oral gavage to feed 4-month-old [*HBx,src,p53^−/−^,RPIA*] fish for one month with the Regorafenib (116 μM), WNK463 (11 μM), Rafoxanide (7.5 μM), and Oligo-fucoidan (6.6 mg/kg). Regorafenib is the second-line clinical treatment for advanced HCC, so it is used as a positive control. The scheme for culturing and oral gavage of HSPR fish is shown in Figure 11a.

We found that the treatments of WNK463+Oligo-focoidan and Rafoxanide+Oligo-fucoidan significantly reduced the expression of proliferation markers (Figure 11b–d). They were even more effective than Regorafenib group. A histopathological examination by H&E staining was used to further verify this finding. The [*HBx,src,p53*^−^*/*^−^*,RPIA*] transgenic fish treated with WNK463+Oligo-focoidan and Rafoxanide+Oligo-fucoidan remarkably reduced the ratio of mitotic figure and karyomegalic cells compared to the hepatocyte from wild-type zebrafish (Figure 11e–g). Senescence is associated with carcinogenesis. During senescence, the cells with senescence-associated secretory phenotype (SASP) secret several inflammatory cytokines and growth factors [44]. We found that WNK463+Oligo-focoidan and Rafoxanide+Oligo-fucoidan significantly reduced the expression of *il1β*, a senescence and inflammation marker (Figure 11h). These results suggest that Oligo-fucoidan might have the ability to attenuate unnecessary inflammation by targeting the WNK1–OSR1 axis.

### 2.9. Oligo-Fucoidan Decreases the Elevated Cell Senescence-Associated β-Galactosidase Activity in Tert Transgenic Fish Treated with WNK1–OSR1 Inhibitors

Cellular senescence is a process associated with chronic inflammation. The senescence-associated β-galactosidase (SA-β-gal), which is overexpressed in senescent cells, has been widely used as a marker of cellular senescence [44]. By using a cellular senescence detection kit, we can detect the senescence-associated β-galactosidase by confocal microscope. We treated tert transgenic fish with oligo-fucoidan, WNK463 without and with Oligo-fucoidan, and Rafoxanide without and with Oligo-fucoidan, and found that Oligo-fucoidan significantly reduced the expression of SA-β-gal in the liver of tert transgenic fish (Figure 12a). After quantification, we found oligo-fucoidan significantly reduced the increased SA-β-gal in tert transgenic fish in the WNK463+OF and Rafox+OF group (Figure 12b). These results may support the idea that Oligo-fucoidan may help to attenuate harmful inflammation while targeting the WNK1–OSR1 axis for cancer therapy.

## 3. Discussion

Angiogenesis is important in both embryonic development and tumor formation, and requires VEGF (vascular endothelial growth factor) binding to VEGFR, tyrosine kinase receptors. VEGF–VEGFR transduces signals leading to endothelial cell proliferation and migration, sprouting and forming a vessel network [45,46,47]. Targeting the VEGF–VEGFR axis including anti-VEGF monoclonal antibodies (e.g., bevacizumab) and VEGF receptor inhibitors (e.g., SU5416) has been developed for anti-tumor induced angiogenesis [48]. Combining atezolizumab (anti-programmed death-ligand 1 [PD-L1]) and bevacizumab has been used as the first-line treatment for advanced HCC [49]. Although the VEGF–VEGFR signaling is common in physiology and pathological angiogenesis, it is very different in structure and function between normal and pathological vessels [50].

Previously, we showed WNK1 is downstream of VEGF-VEGFR in embryonic angiogenesis [24], and endothelial WNK1-mediated tumor-induced angiogenesis promotes tumor proliferation and metastasis in zebrafish [15]. In this study, we demonstrated that the downstream effectors for WNK1 are different between embryonic angiogenesis and tumor-induced angiogenesis. The expression of *wnk1a* and *osr1a* is upregulated at 36–48 hpf during embryonic angiogenesis, while in tumor-induced angiogenesis, *wnk1a* and *osr1b* are upregulated and *ppp2r1ba/ppp2r1bb* are downregulated at 3dpi with tumor-induced angiogenesis promoting tumor cell proliferation and migration. The different downstream effectors might lead to distinct organization of vessels and discordant tasks between physiologic and pathological angiogenesis, and targeting this unique function of tumor-induced angiogenesis might be an effective anti-tumor strategy.

Transient receptor potential cation channel subfamily V member 4 (TRPV4) ubiquitously expresses in many organs and tissues [51]. TRPV4 mediates the reorientation of endothelial cells to integrin signaling, which is the first step of neoangiogenesis [52]; however, the expression of TRPV4 was suppressed in tumor endothelial cells resulting in abnormal angiogenesis via increased Rho activity, and the combination therapy of Rho kinase inhibitor and anti-cancer drug can significantly reduce tumor growth [53]. This indicates that targeting the distinct properties of tumor-induced angiogenesis might be a potential anticancer therapy. It has been reported that the expression and activity of TRPV4 can be inhibited by WNK1 in the kidney [54]. It may be possible that the overexpression of *wnk1a* and *osr1b* in endothelial cells may suppress TRPV4 expression in tumor induced angiogenesis.

In this study, we demonstrated that endothelial cells promote hepatoma cell migration via the WNK1–OSR1 axis. Knockdown of WNK1 in endothelial cells diminishes hepatoma cell migration, and it can be rescued by OSR1 overexpression. We speculate that the WNK1–OSR1 axis promotes tumor migration through epithelial–mesenchymal transition (EMT) which is a vital process for cancer cell migration and invasion. In the process of EMT, the cell–cell adherence reduces, and motile potential increases [55]. SLUG, also known as SNAIL2, is a zinc finger protein that triggers the initial step of EMT [56]. In accordance with our results, the knockdown of WNK1-OSR1 decreased the migration of HUVEC cells in concert with a decline in SLUG [57]. SPAK activates SNAIL by phosphorylating SNAIL and preventing SNAIL from degrading [19]. HUVEC cells can secrete high levels of TGF-β that can induce tumor cells to undergo EMT [58,59]. We speculate that endothelial WNK1–OSR1 may activate TGF-β to be secreted into the medium to stimulate hepatoma cell migration.

It has been reported that OSR1 interacts with Smad 2/3 to stimulate its translocation to the nucleus, enhancing the transcription of EMT factors and TGF-β. Through the autocrine of TGF-β, tumor cells maintain an environment conducive to cancer growth and migration [13]. The ion influx accompanied by water flux leads to volume gain at the cell front and volume loss at the rear part of migrating cells [25]. Elevated expression of NKCC1, downstream of the WNK1–OSR1 axis, enhanced the migration and invasion ability [18]. TMZ treatment activates the WNK1/OSR1/NKCC1 pathway, thus increasing the glioma migration [17]. Under hypoxia, tumor cells secrete VEGF, which acts in both autocrine and paracrine manners to affect tumor cells. VEGF promotes surrounding endothelial cell growth and triggers angiogenesis. Our previous results indicated that WNK1 is downstream of the VEGF–VEGFR–PI3K/AKT axis in affecting angiogenesis since the mRNA of *wnk1a* and *wnk1b* can partially rescue the effect of *flk1*/*vegfr2* knockdown, but the AKT/PI3K phosphorylation site mutation of wnk1a cannot rescue it [24]. It has been reported that the formation of the lumen was increased by incubating time and the number of exosomes secreted by hepatoma cells. This indicated that the exosomes containing RNA and miRNA may have a positive influence on the lumen formation of HUVECs [60].

In this study, we found that in both xenotransplantation and the *tert* transgenic fish HCC model, *ppp2r1bb* is suppressed, suggesting *ppp2r1bb* might be a tumor suppressor. Moreover, overexpression of PPP2R1A can eliminate the increased migration ability induced by WNK1 overexpression in hepatoma cells, supporting the tumor suppressor role of PPP2R1A. We hypothesize that two effects were involved: FOXOs are considered tumor suppressors due to their ability in regulating cell cycle arrest, apoptosis, differentiation, DNA damage repair, senescence, and scavenging of reactive oxygen species [61]. Direct binding of WNK1 and PPP2R1A promotes functional PP2A activity. PP2A negatively regulates AKT, decreasing the active form of AKT. Thus, AKT cannot translocate to the nucleus to trigger the nuclear exclusion of FOXO1 by phosphorylating it. FOXO1 transcription was maintained to have a tumor-suppressing effect [23]. In other studies, the phosphorylation of Akt1/3, VEGF signaling, and WNK1 that are involved in angiogenesis were increased in mutant PPP2R1A expressing GIST cells detected by phospho-kinase array. The phosphatase 2A intends to dephosphorylate WNK1 but fails to do so due to the mutation in its subunit-PPP2R1A [41]. The mutation of PPP2R1A was discovered in many cancers such as endometrial cancer, ovarian and uterine carcinomas, breast cancer, and gastrointestinal stromal tumors [41,62,63]. Although direct evidence is lacking that there is a mutation of PPP2R1A in HCC, it is possible that mutation of PPP2R1A affects the activity of WNK1 in HCC.

A previous report indicated that loss of WNK appears to be pro-inflammatory [30]. It is anticipated that these effects would be detrimental because WNK1 has been reported to antagonize inflammasomes by suppressing NLRP3 activation [30], and thus targeting WNK1 might induce inflammation. When considering the WNK pathway as a therapeutic target it should be noted that it may induce undesired excessive inflammatory responses. Since oligo-fucoidan is well known for its immune modulation function, we combined oligo-fucoidan to prevent unwanted inflammation by targeting WNK1–OSR1. Our results reveal the advantage of combination therapy with WNK1–OSR1 inhibitor and oligo-fucoidan, which could be a potential therapeutic approach for advanced HCC.

To understand the mechanism of oligo-fucoidan in the anti-HCC effect, we hypothesize that oligo-fucoidan exhibits an anti-inflammatory effect, thus reducing cancer formation. From the specimens of human cancer patients, signs of senescence, including oxidative damage accumulation and TP53 upregulation, were detected in normal tissues adjacent to the tumors. Senescence-associated secretory phenotype (SASP)-mediated propagation of senescence and abnormal proliferation in the neighborhood may be a new model of heterogeneous tumor generation, and NAC has been proven to reduce the SASP and tumor formation [64]. We investigated the expression of inflammation/senescence in transgenic fish with HCC before and after WNK1–OSR1 inhibitors plus oligo-fucoidan treatment and confirmed that the effect of oligo-fucoidan can avoid undesired inflammation. We examined the senescence phenotype by checking the senescence-associated β-galactosidase (SA-β-gal) accumulation and found the SA-β-gal accumulation is indeed increased in *tert* transgenic fish, and oligo-fucoidan can reduce the SA-β-gal accumulation in *tert* transgenic fish treated with WNK1–OSR1 inhibitors, thus further reducing cancer formation.

To sum up, we reveal the WNK1 downstream effectors involved in tumor-induced angiogenesis and tumor migration, and PPP2R1A acts as a tumor suppressor. The WNK1–OSR1 axis mediates tumor-induced angiogenesis and migration perhaps through many other pathways, such as TGF-β, VEGF, and ion transporters. Although WNK1–OSR1 inhibitors might cause an unwanted inflammation response and could be toxic, our combination therapy approach by integrating WNK1–OSR1 inhibitors with oligo-fucoidan (an anti-inflammatory agent) might serve as a promising novel anti-cancer therapeutic.

## 4. Materials and Methods

### 4.1. Zebrafish Husbandry

AB wild-type, Tg(fli1:EGFP), Tg(fabp10a:HBx,src,RPIA;myl7:EGFP)xtp53^zdf1/zdf^, Tg(fli1:wnk1a;myl7:EGFP), Tg(fli1:EGFP)xTg(fli1:CreERT2;myl7:EGFP)xTg(loxP-wnk1a-DsRed-loxP), Tg(fli1:EGFP)xTg(fli1:wnk1a), Tg(fabp10a:tert;myl7:EGFP)xtp53^zdf1/zdf^, and Tg(fabp10a:tert;myl7:EGFP) were used in this study, and their phenotypes are described in Table 1.

To knockout the *wnk1a*, we treated the Tg(fli1:EGFP)xTg(fli1:CreERT2;myl7:EGFP)xTg(loxP-wnk1a-DsRed-loxP) transgenic fish with 1 µM of RU486 from 0 to 5 dpf. The zebrafish were maintained in the Zebrafish Core Facility at National Health Research Institute (NHRI) where the temperature is kept at 28 °C. Zebrafish were cultured in continuous flow with constant aeration on a 14:10 h light–dark cycle. All zebrafish experiments were approved by the Institution Animal Care and Use Committee (IACUC) of the NHRI (protocol No. NHRI-IACUS-108035-A).

### 4.2. Embryo Collection

One night before collecting embryos for injection or another purpose, the male and female fish were put in a mating tank with a divider to separate the fish. The divider was removed and changed to a new tank with fresh water before 9:30 a.m., for the fish to mate. The embryos were collected for microinjection after being rinsed with water to avoid excrement pollution, and put in E3 buffer in the incubator at 28 °C.

### 4.3. Cell Culture

In this study, we investigated the WNK1 axis in endothelial cells interacting with the WNK1 axis in hepatoma cells, promoting hepatoma cell migration, by a series of in vitro co-culture and transwell assay experiments. HCC cells (HepG2 and Hep3B) and human umbilical vein endothelial cells (HUVECs) were used. All the cell lines were purchased from Bioresource Collection and Research Center (BCRC). HepG2 and Hep3B were cultured in Dulbecco’s modified Eagle’s medium (C-DMEM) (Gibco, Billings, MT, USA) supplemented with 10% fetal bovine serum (FBS) (Gibco, Billings, MT, USA) and 1% penicillin-streptomycin (Gibco, Billings, MT, USA). The HUVECs are primary cells, and we used passage 3 and culture in EGM-2 BULLET KIT (LONZA, Basel, Switzerland) with an additional 1% penicillin-streptomycin (Gibco, Billings, MT, USA). All cells were maintained at 37 °C in an incubator with 5% CO_2_.

### 4.4. Xenotransplantation

For investigating the WNK1 axis in tumor-induced angiogenesis, Hep3B hepatoma cells were injected into two days post-fertilization (dpf) embryos of Tg(fli1:EGFP) fish using xenotransplantation. At 1 dpf, the embryos were dechorionated with pronase (Sigma-Aldrich Inc. Burlington, MA, USA) before noon and maintained in 1× PTU/E3 buffer. The 2 dpf embryos were anesthetized and arrayed on a 2% agarose plate under a microscope. The Hep3B hepatoma cells were labeled with Celltrace yellow dye, washed, and resuspended as single-cell suspensions, of which 4.6 nL containing 200 cells was injected into each embryo at the middle of the yolk sac. After xenograft, the embryos were incubated in a RI-80 low-temperature incubator, which was programmed to gradually increase the temperature (one °C every six hours) from 28 °C to 37 °C over two days, and subsequently to maintain it at 37 °C until the end of this experiment.

To observe the tumor-induced angiogenesis, tumor cell proliferation, and migration, the embryos were anesthetized from 1 to 3 days post-injection (dpi), each embryo was placed in a well of 96-well plate, and images were taken under a Leica DM IRB DMIRB Inverted Fluorescence Microscope. Endothelial cells expressing green fluorescence can be used to observe angiogenesis, while hepatoma cells labeled with Celltrace yellow dye irradiated red fluorescence were used to monitor tumor cell proliferation and migration, analyzed by ImageJ software. Furthermore, 30 embryos from 1–3 dpi were collected to quantify the expression of WNK1, WNK1 downstream effectors, and PP2A subunits from both humans and zebrafish by QPCR. Primer sequences were designed to detect specific genes as listed in Table 2.

### 4.5. Embryonic Angiogenesis

To explore the WNK1 axis in embryonic angiogenesis, the embryos of AB wild-type were collected at 24, 36, and 48 h post-fertilization for measuring RNA expression levels. For each time point, 30 embryos were collected, lysed, and RNA was extracted for QPCR to detect gene expression in zebrafish WNK1 and its downstream effectors, along with pp2a subunits. The information on primers is listed in Table 2.

### 4.6. RNA Extraction

Total RNA of fish specimens was extracted with a NucleoSpin^®^ RNA kit (MACHEREY-NAGEL, Bethlelm, PA, USA). About 30 mg of fish tissue was collected and suspended in 350 μL buffer RA1 with 3.5 μL β-mercaptoethanol (Sigma-Aldrich Inc., Burlington, MA, USA) mixture. Then the sample was homogenized by Bullet Blender Tissue Homogenizer with RNase Free Zirconium Oxide Beads 0.5 mm (ZrOB05, Next Advance Inc., NY, USA). To reduce viscosity and clear the lysate, it was filtrated through NucleoSpin^®^ Filter by centrifuging for 1 min at 11,000× *g*. Then, 350 μL of 70% ethanol was added and mixed well. The lysate was loaded to NucleoSpin^®^ RNA Column and centrifuged for 30 s at 11,000× *g*. Next, 350 μL of MDB (membrane desalting buffer) was loaded and centrifuged at 11,000× *g* for 1 min to dry the membrane. To digest genomic DNA, 95 μL of rDNase reaction mixture was applied onto the center of the silica membrane of the column. This mixture was prepared by adding 10 μL of reconstituted DNase to 90 μL of reaction buffer for each isolation. The column was incubated at room temperature for at least 15 min, but no more than 1 h. After DNA digestion, the silica membrane was washed three times. To inactivate the rDNase, 200 μL of RAW2 buffer was added to the column and centrifuged for 30 s at 11,000× *g*. To wash the column, 600 μL of RAW3 buffer was loaded in the column and centrifuged for 30 s at 11,000× *g*. To wash away all the residual buffers from previous steps, 250 μL of RAW3 buffer was applied to the column and centrifuged for 30 s at 11,000× *g*. To dry the silica membrane thoroughly, it was centrifuged for another 2 min at 11,000× *g*. The columns were transferred to new RNase-free 1.5 mL tubes and RNA samples were eluted in 40 μL of RNase-free water and centrifuged at 11,000× *g* for 1 min. The concentration of all RNA samples was quantified by a Thermo NanoDrop ND-1000 UV-Vis Spectrophotometer and then stored at −80 °C.

### 4.7. Reverse-Transcription and Quantitative Polymerase Chain Reaction (QPCR)

The complementary DNA (cDNA) was reverse-transcribed from total RNA by iScript^TM^ Synthesis Kit (BIO-RAD, Hercules, CA, USA). The components are listed as Table 3 below:

Reaction was set at 25 °C for 5 min, 46 °C for 20 min, 95 °C for 1 min, and soak at 4 °C.

After the RT reaction, cDNA was diluted (100-fold to 20-fold) for QPCR. The component and QPCR program were as shown below. The sample and buffer were loaded in a 384-well plate (Labcon, CA, USA) and sealed with Optical Adhesive Covers (Applied Biosystems, Waltham, MA, USA). SYBR Green fluorescence was detected by QuantStudio™ 5 Real-Time PCR Systems. The components are listed as Table 4 below:

The PCR program is described as Table 5 followed:

All Ct volumes of target genes were normalized to internal control actin, and the ΔCt = (Ct _target_ − Ct _actin_) was obtained. After normalization, the expression ratio between experimental and control groups was calculated using the comparative Ct method. The relative expression ratio (fold change) was calculated based on ΔΔCt = ΔCt _treatment_ − ΔCt_control_, and the fold changes are equal to 2^−ΔΔCt^.

All experiments were performed in triplicate and at least three independent samples were used for QPCR. The mean value and the standard error were calculated and incorporated into the presented data as medians ± standard error.

### 4.8. Transfection

To knockdown the expression of human WNK1, two shWNK1 were ordered from Academia Sinica RNAi Core. The clone IDs of shWNK1-1 and shWNK1-2 are TRCN0000000919 and TRCN0000000920 respectively. The sequence of the shRNA is the following:

shWNK1-1:5′-CCGG-CCGCGATCTTAAATGTGACAA-CTCGAG-TTGTCACATTTAAGATCGCGG-TTTTT-3′

shWNK1-2:5′-CCGG-GCGTAGTTTCAAGTATCACAA-CTCGAG-TTGTGATACTTGAAACTACGC-TTTTT-3′

Transfection of HUVECs was performed using Lipofectamine 2000 (Invitrogen, Waltham, Massachusetts, USA). The day before transfection, HUVECs were seeded in the lower chamber of a 24-well transwell plate for transwell assay with a density of 1 × 10^4^, or seeded in a 6-well plate for Western blot at a density of 1 × 10^5^. The cells grow into 80–90% confluent on the day of transfection, and the growth medium is then replaced with a serum-free growth medium for transfection. We used Lipofectamine 2000 with 0.8 μg DNA and 2 μL Lipo 2000 for 24-well plate, and 4 μg DNA and 10 μL Lipo 2000 for a 6-well plate. DNA and Lipo 2000 was diluted into Opti-MEM Medium (Gibco, Montana, USA), and then the diluted DNA and Lipo 2000 were combined together, mixed well by pipetting, incubated at room temperature for 20 min, and then added to the cells. The time for transfection was 4 h.

Transfection of HepG2 was performed in a 6 cm dish with 5.5 μg of DNA, 8.5 μL of Lipo 3000, and 11 μL of P3000 using a Lipofectamine 3000 kit (Invitrogen, Waltham, Massachusetts, USA). The day before transfection, HepG2 cells were counted to appropriate density. The cells would be 60–70% confluent on the day of transfection, and then the growth medium was replaced with a serum-free growth medium for transfection. DNA was diluted in Opti-MEM Medium, P3000 and mixed well. L3000 was also diluted into Opti-MEM Medium. Then, the diluted DNA and L3000 were combined, mixed well by pipetting, incubated at room temperature for 20 min, and then added to the cells. The cells were incubated in a humidified atmosphere with 5% CO_2_ at 37 °C for 6 h, the medium was changed back to a growth medium and recover for another 24 h, and then seeded in the top well of transwell plate for migration assay.

### 4.9. Transwell Migration Assay

The HUVECs were seeded in the lower chamber with 10% FBS growth medium before the HepG2 hepatoma cell seeding at the density of 5 × 10^5^ in the upper chamber with 100 μL FBS free growth medium. The cells were incubated in a humidified atmosphere with 5% CO_2_ at 37 °C for 18 h. The medium was removed from the upper and lower chambers. The upper chamber was rinsed with PBS twice, and the cells were fixed with 3.7% formaldehyde for 5 min. The transwell insert was washed with PBS twice again. An amount of 100% methanol was added for 20 min to permeabilize, then rinsed with PBS twice. Cells were stained with 0.5% crystal violet (in 2% ethanol) for 10–15 min, and then the insert was washed with PBS twice to remove unbound crystal violet. The cells on the upper side of transwell membranes were gently removed using moistened cotton swabs. Finally, the membrane was air-dried and pictures taken under a Leica DMIL microscope. All the data of the transwell migration assay were analyzed by ImageJ. All experiments were performed in triplicate. The mean value and the standard error were calculated and incorporated into the presented data as median ± standard error.

### 4.10. Western Blot Analysis

To determine effective knockdown of WNK1 or overexpression of WNK1, OSR1, and PPP2R1A through transfection and the impact on downstream mediators, we performed Western blot analysis. Cells were lysed in RIPA buffer (25 mM Tris-HCl pH 7.6, 150 mM NaCl, 5 mM EDTA, 1% NP-40 or 1% Triton X-100, 1% sodium deoxycholate, 0.1% SDS) supplemented with protease inhibitor cocktail (MedChemExpress, Monmouth Junction, NJ, USA) and centrifuged at 13,000× *g* for 15 min at 4 °C. The supernatant was collected, and protein concentration was quantified using a Pierce^TM^ BCA Protein assay Kit (Thermo Scientific, MA, USA) following the company’s instructions. A total of 40 μg protein was resolved by 4–20% SurePAGE, Bis-Tris, 10 cm × 8cm pre-cast gel (GenScript, Piscataway, NJ, USA) at 80 Volt for 2 to 2.5 h, and transferred onto PVDF membranes (Immobilon, MERCK, Kenilworth, NJ, USA) at 90 Volt for 2.5 h. The membrane was then blocked by HyBlock 1min Blocking Buffer (GOAL Bio, Taipei, Taiwan) for 5 min on a shaker. After blocking, the membrane was incubated with a specific primary antibody at 4 °C overnight, and washed by TBST buffer three times for 20 min each. After that, the membrane was incubated with a secondary antibody at 4 °C for 1 h and washed again. Chemiluminescence was detected with T-Pro LumiLong Plus Chemiluminescence Detection Kit (T- Pro Biotechnology, New Taipei County, Taiwan) by ChemiDoc^TM^ MP Image System (BIO-RAD, Hercules, CA, USA). All the data were analyzed by ImageJ. The primary antibodies were anti-WNK1 (#4979, Cell signaling, Danvers, MA, USA), anti-OSR1 (#3729, Cell signaling, MA, USA), anti-PPP2AA (#2039, Cell signaling, MA, USA), and anti-β-actin (GTX109639, GeneTex, Hsinchu, Taiwan). The secondary antibody was Anti-rabbit IgG, HRP-linked antibody (#7074, Cell signaling, MA, USA).

### 4.11. Oral Gavage

Regorafenib, WNK463 (MedChemExpress, Monmouth Junction, NJ, USA), Rafoxanide (MedChemExpress, Monmouth Junction, NJ, USA), and oligo-fucoidan were administered via oral gavage to Tg(fabp10a:HBx,src,RPIA;myl7:EGFP) × tp53^zdf1/zdf^ to examine the anti-cancer effect in the HCC fish model. These fish were fed with a normal diet from 5 dpf to 5 months old. At four months of age, the fish was divided into four groups in 3 L tanks: #1: no drug treatment as negative control; #2: 5 μL containing 116 μM Regorafenib as positive control; #3: 5 μL solution containing 11 μM WNK463 plus 5 μL solution containing 6.6 mg/kg oligo-fucoidan; #4: 5 μL solution containing 7.5 μM Rafoxanide plus another 5 μL solution containing 6.6 mg/kg oligo-fucoidan. The fish was anesthetized and put into a wet sponge preventing moving. The medicine solution was fed with microliter syringes (HAMILTON, Reno, NV, USA) and FTP-22–25 plastic feeding tubes (INSTECH, Plymouth Meeting, PA, USA). After feeding, the fish were put back into the 3 L tank for recovery. The medicine was given twice a week continuing for one month. The fish were sacrificed for examination at five months of age.

### 4.12. Hematoxylin and Eosin Stain

The *tert* and *tertxp53^−/−^* transgenic fish larvae and [*HBx,src,p53^−/−^,RPIA*] transgenic fish adult liver tissues were fixed in 10% formaldehyde and changed to 75% ethanol. Hematoxylin and eosin stains were performed by the pathology core lab (NHRI, Miaoli, Taiwan).

### 4.13. Detection of Senescence-Associated β-galactosidase (SA-β-gal) Accumulation

The Cellular Senescence Detection Kit—SPiDER-βGal (DOJINDO, Kumamoto, Japan, SG03) was used to detect the SA-β-gal-positive senescent cells according to the protocol described previously [64]. In detail, 5 larvae were put into each well of a 24-well plate, and washed with 500 µL of PBS once, and then larvae were fixed by adding 500 µL of 4% paraformaldehyde in PBS (4% PFA) for 10 min at RT. The supernatant was removed, and the larvae were washed with 500 µL of PBS three times; then, 500 µL of SPiDER-βGal working solution in McIlvaine buffer (pH 6.0, dilution 1/1000) was added to the well for 30 min at 37 °C. After removing the supernatant, the larvae were washed with 500 µL of PBS twice, and then the larvae were observed, and the images were taken by confocal microscope. The quantification of SA-β-gal accumulation was performed by using a Cellular Senescence Plate Assay Kit—SPiDER-βGal (DOJINDO, Kumamoto, Japan, SG05). In detail, five larvae per well were washed with 500 µL of PBS once, then 400 µL of lysis buffer was added to each well and the plate was incubated at room temperature for 10 min; 50 µL of lysate solution was transferred to each well of a 96-well black plate, and 50 µL of SPiDER-βGal working solution was added to each well and incubated at 37 °C for 30 min. Then, 100 µL of stop solution was added to each well and the fluorescence was measured using a fluorometer (excitation: 500–540 nm, emission: 540–580 nm). Triplicates were done for each sample, and normalized to the no-reagent control to remove the background signal, and fold changes were calculated using WT as 1.

### 4.14. Statistical Analysis

All statistical analyses except Figure 9 were conducted using one-way ANOVA analysis followed by multiple analysis of variance, used to identify the statistical significance between groups. Figure 9 used the unpaired Student’s *t*-test (two-tailed test) for statistical analysis. All figures were drawn using Prism 7 and 9. The *p*-value is shown as: *: *p* ≦ 0.05, **: *p* ≦ 0.01, ***: *p* ≦ 0.001, and ****: *p* ≦ 0.0001.

## Figures and Tables

**Figure 1 ijms-23-12100-f001:**
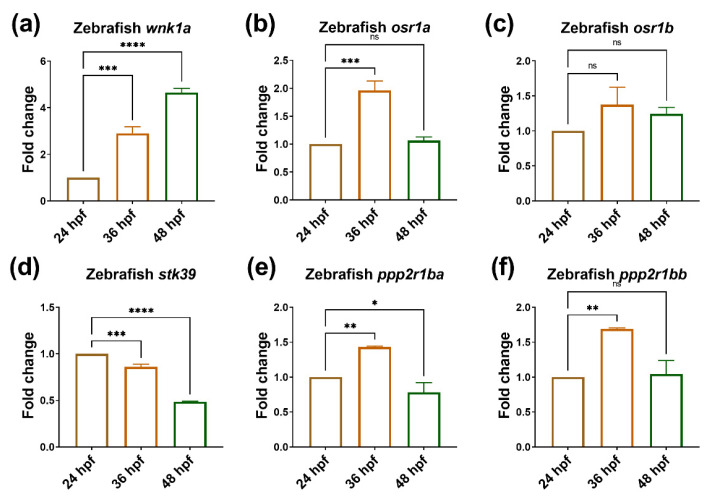
During embryonic angiogenesis, *wnk1a* and *osr1b* are upregulated. The expression of *wnk1* and its downstream effectors during embryonic angiogenesis. (**a**) *wnk1a*, (**b**) *osr1a*, (**c**) *osr1b*, (**d**) *stk39*, (**e**) *ppp2r1ba* and (**f**) *ppp2r1bb* in zebrafish embryos stages from 24 h post-fertilization (hpf) to 48 hpf during embryonic angiogenesis. For each experiment, we performed three replicates, and for each qPCR, there were triplicates for each sample. Thirty larvae were included in each time point. One-way ANOVA analysis followed by multiple analysis was used to identify the statistical significance between groups; *: *p* ≦ 0.05; **: *p* ≦ 0.01; ***: *p* ≦ 0.001; ****: *p* ≦ 0.0001; ns: no-significance.

**Figure 2 ijms-23-12100-f002:**
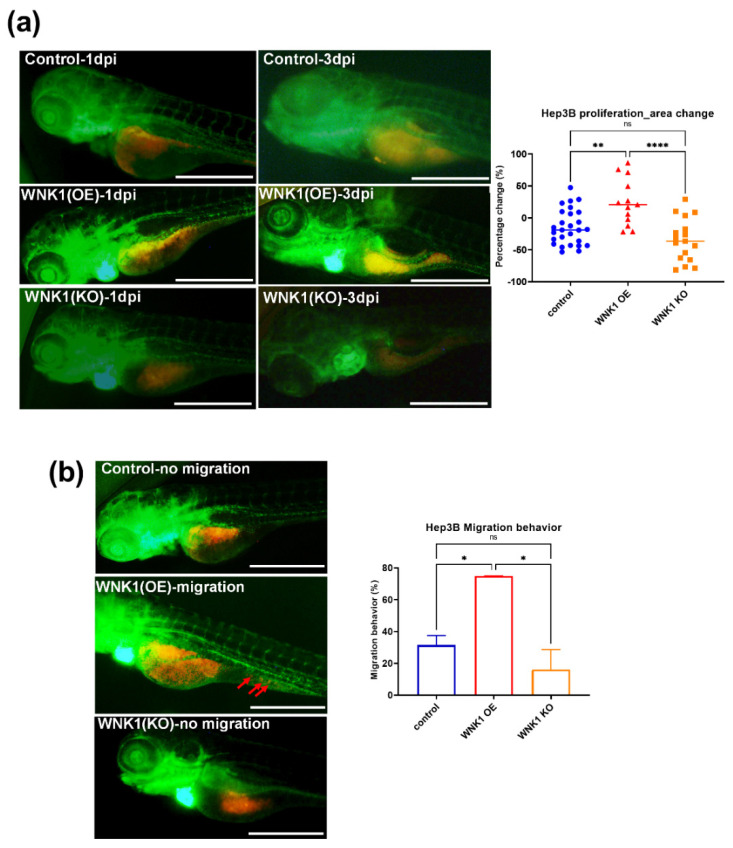
Tumor-induced angiogenesis promotes hepatoma cell proliferation and migration. (**a**) The representative stacked images of 1 day post injection (dpi) and 3 dpi after red fluorescence-labeled Hep3B cells were injected into green fluorescence blood vessel embryos of Tg(fli1:EGFP)-control, Tg(fli1:EGFP)xTg(fli1:wnk1a)-WNK1(OE), and Tg(fli1:EGFP)xTg(fli1:CreER^T2^;myl7:EGFP)xTg(loxP-wnk1a-DsRed-loxP) after RU486 treated for knockout wnk1a-(WNK1(KO)). The images showed the tumor-induced angiogenesis with ectopic vessels surrounding the Hep3B cells. Quantification of area change comparing 3 dpi versus 1 dpi in control, WNK1 OE, or WNK1 KO embryos. (**b**) The representative images for control, WNK1 OE, or WNK1 KO embryos, red arrows indicate the migrated cells in WNK1(OE). Quantification of migration behavior comparing 3 dpi versus 1 dpi after red fluorescence-labeled Hep3B cells were injected into control, WNK1(OE), and WNK1(KO) embryos. We performed three replicates, and ten to thirty embryos were included in each experiment. One-way ANOVA analysis followed by multiple analysis was used to identify the statistical significance between groups; *: *p* ≦ 0.05; **: *p* ≦ 0.01; ****: *p* ≦ 0.0001; ns: no-significance. The scale bar is 500 µm in length.

**Figure 3 ijms-23-12100-f003:**
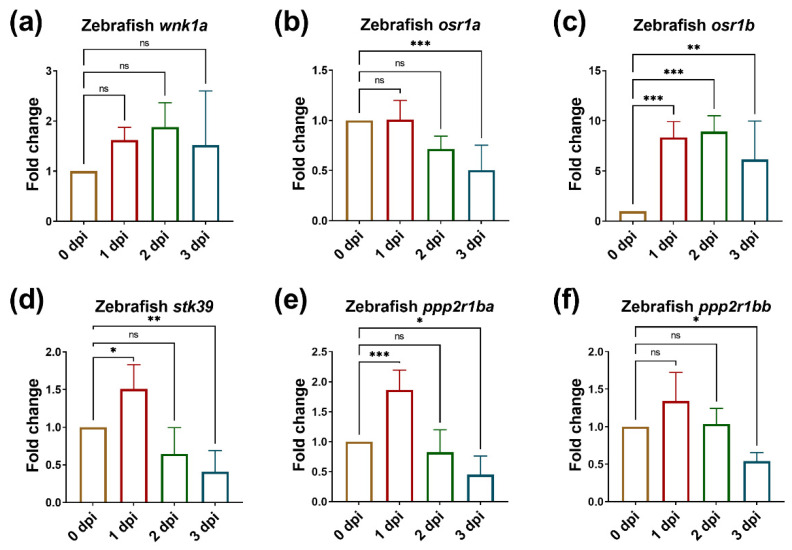
The expressions of *wnk1a* and *osr1b* are upregulated in endothelial cells during tumor-induced angiogenesis. The expression of *wnk1a* and its downstream effectors during tumor-induced angiogenesis. (**a**) *wnk1a*, (**b**) *osr1a*, (**c**) *osr1b*, (**d**) *stk39*, (**e**) *ppp2r1ba* and (**f**) *ppp2r1bb* in zebrafish embryos from 0 dpi to 3 dpi during tumor-induced angiogenesis. We performed three to four replicates, and for each qPCR, there were triplicates for each sample. Thirty embryos were included in each experiment. One-way ANOVA analysis followed by multiple analysis was used to identify the statistical significance between groups. *: *p* ≦ 0.05; **: *p* ≦ 0.01; ***: *p* ≦ 0.001; ns: no-significance.

**Figure 4 ijms-23-12100-f004:**
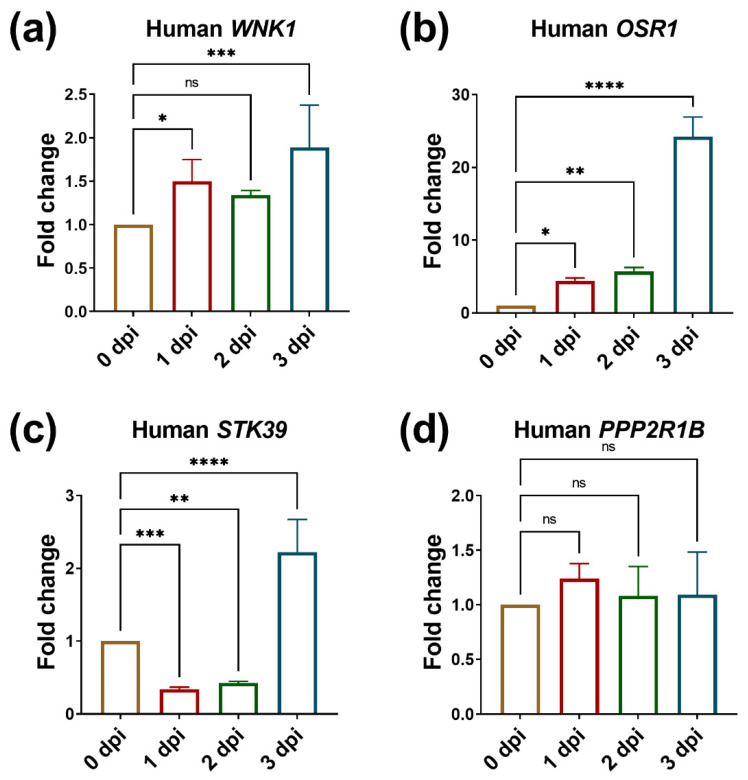
The expression of *WNK1* and *OSR1* is upregulated in hepatoma cells during tumor-induced angiogenesis promoting hepatoma cell proliferation. The expression of *WNK1* and its downstream effectors in hepatoma cells during tumor induced angiogenesis. (**a**) *WNK1*, (**b**) *OSR1*, (**c**) *STK39*, (**d**) *PPP2R1B* in zebrafish embryos carrying hepatoma cells from 0 dpi to 3 dpi during tumor-induced angiogenesis. We performed three to four replicates, and for each qPCR, there were triplicates for each sample. Thirty embryos were included in each experiment. One-way ANOVA analysis followed by multiple analysis was used to identify the statistical significance between groups. *: *p* ≦ 0.05; **: *p* ≦ 0.01; ***: *p* ≦ 0.001; ****: *p* ≦ 0.0001; ns: no-significance.

**Figure 5 ijms-23-12100-f005:**
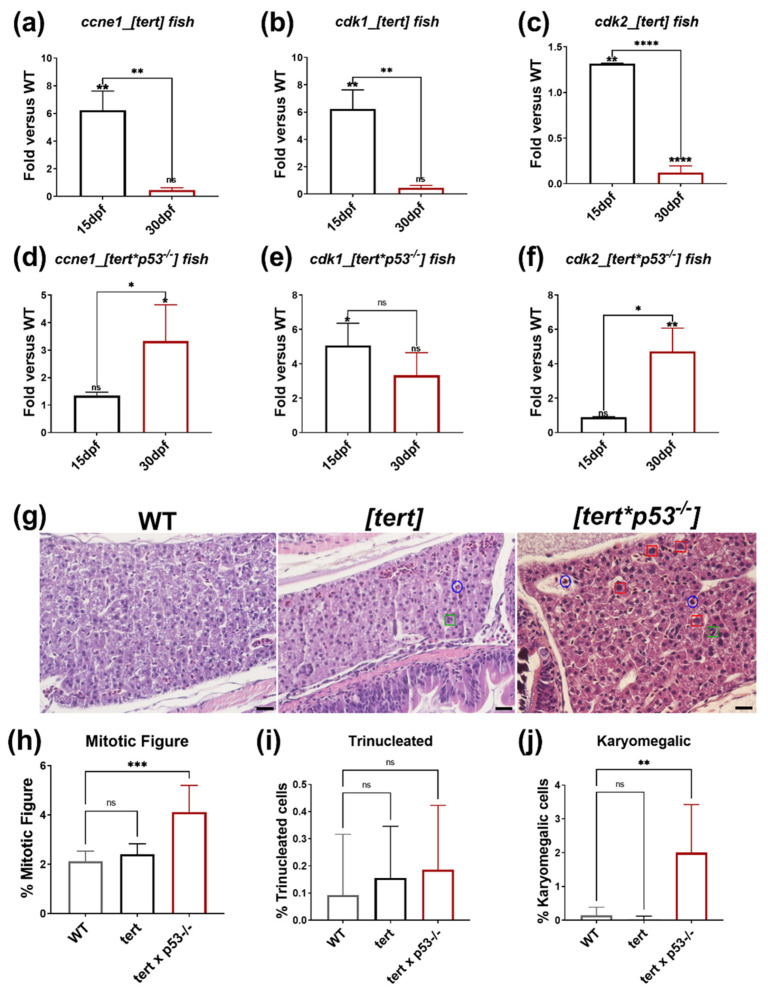
The expression of cell proliferation markers and histopathological features from H&E stain in [*tert*] and [*tert x p53^−/−^*] transgenic fish. The expression of proliferation marker in [*tert*] fish at 15 dpf and 30 dpf (**a**) *ccne1*, (**b**) *cdk1*, and (**c**) *cdk2*, and in [*tert x p53^−/−^*] fish at 15 dpf and 30 dpf (**d**) *ccne1*, (**e**) *cdk1*, and (**f**) *cdk2*. *: *p* ≦ 0.05; **: *p* ≦ 0.01; ***: *p* ≦ 0.001. (**g**) The representative images of H&E stain of WT, [*tert*], and [*tert x p53^−/−^*] fish at 30 dpf. The blue circle indicates the mitotic features, the green square points out the trinucleated hepatocytes, the red square denotes the karyomegalic cells. The scale bar is 20 µm in length. The quantification of histopathological features from H&E stain: (**h**) mitotic figures, (**i**) trinucleated hepatocytes, (**j**) karyomegalic cells are increased in WT, [*tert*], and [*tert x p53^−/−^*] fish at 30 dpf. We performed three replicates, and for each qPCR, there were triplicates for each sample. Ten embryos were included in each experiment. One-way ANOVA analysis followed by multiple analysis was used to identify the statistical significance between groups. *: *p* ≦ 0.05; **: *p* ≦ 0.01; ***: *p* ≦ 0.001; ****: *p* ≦ 0.0001; ns: no-significance.

**Figure 6 ijms-23-12100-f006:**
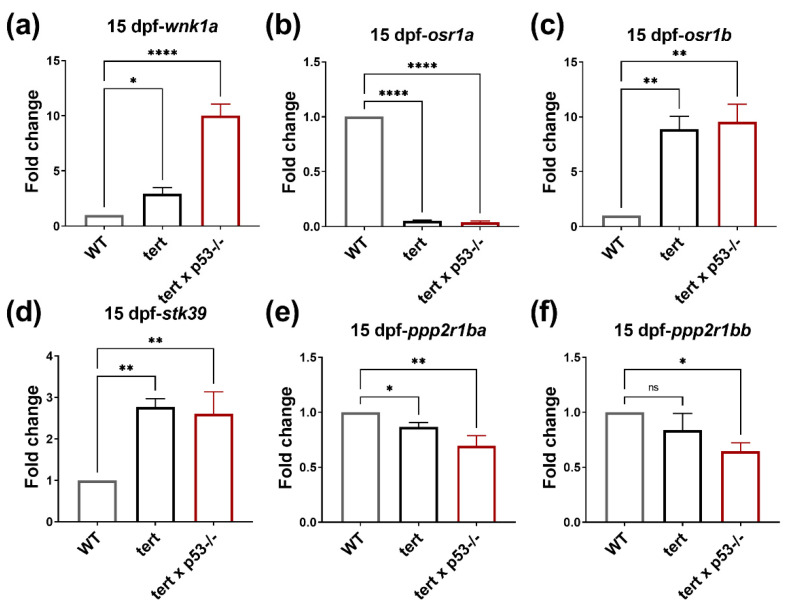
Upregulated *wnk1a*, *stk39*, and *osr1b*, and downregulated *ppp2r1ba* and *ppp2r1bb* in 15 dpf [*tert*] and [*tert x p53^−/−^*] transgenic fish. The expression of *wnk1a* and its downstream effectors in WT, [*tert*], and [*tert x p53^−/−^*] fish at 15 dpf. (**a**) *wnk1a*, (**b**) *osr1a*, (**c**) *osr1b*, (**d**) *stk39*, (**e**) *ppp2r1ba*, and (**f**) *ppp2r1bb* in 15 dpf [*tert*] and [*tert x p53^−/−^*] transgenic fish compared to WT. We performed three replicates, and for each qPCR, there were triplicates for each sample. Ten larvae were included in each experiment. One-way ANOVA analysis followed by multiple analysis was used to identify the statistical significance between groups. *: *p* ≦ 0.05; **: *p* ≦ 0.01; ****: *p* ≦ 0.0001; ns: no-significance.

**Figure 7 ijms-23-12100-f007:**
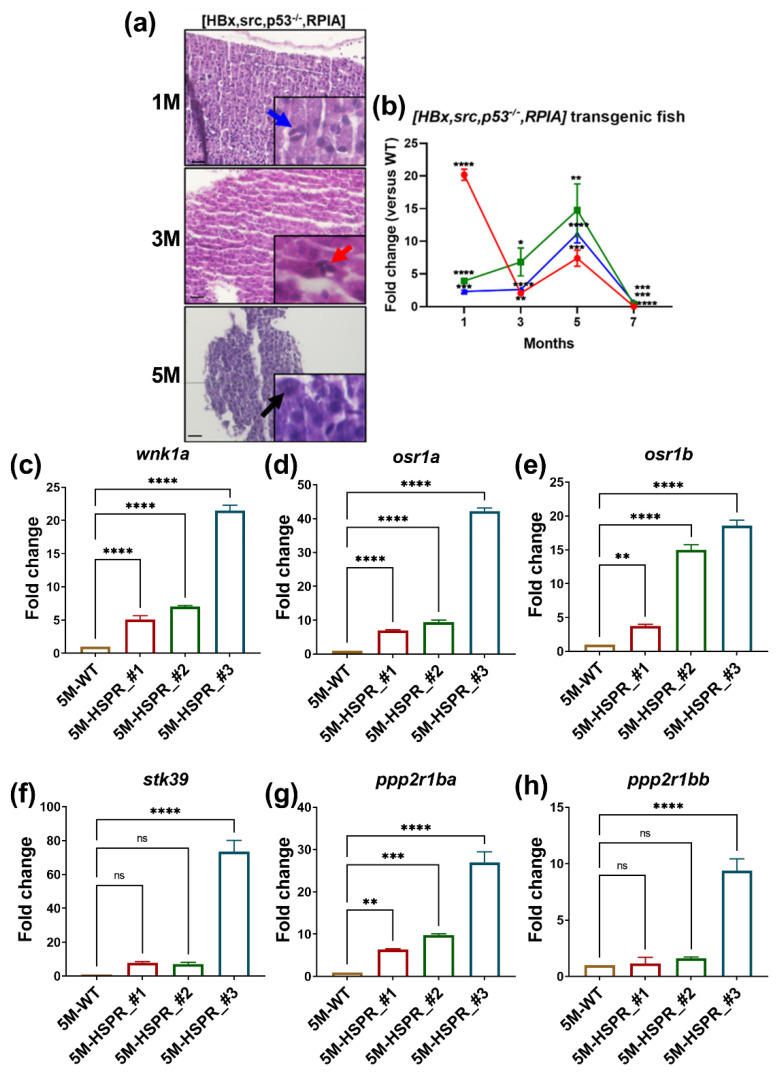
H&E stain and QPCR analysis reveal that [*HBx,src,p53^−/−^,RPIA*] transgenic fish developed HCC at 5 months of age. (**a**) The representative images of H&E staining of 1, 3, 5M [*HBx,src,p53^−/−^,RPIA*] transgenic fish. The scale bar is 20 µm in length. (**b**) The expression of proliferation marker-*ccne1*, *cdk1*, and *cdk2* of 1, 3, 5M [*HBx,src,p53^−/−^,RPIA*] transgenic fish. The expression of *wnk1a* and its downstream effectors in WT and [*HBx,src,p53-/-,RPIA*] transgenic fish at 5 months of age. (**c**) *wnk1a*, (**d**) *osr1a*, (**e**) *osr1b*, (**f**) *stk39*, (**g**) *ppp2r1ba*, and (**h**) *ppp2r1bb* at 5 months of #2, #3, #4 independent [*HBx,src,p53^−/−^,RPIA*] transgenic fish compared to WT. We performed three replicates, and for each qPCR, there were triplicates for each sample. Multiple adult fish were included in each experiment. One-way ANOVA analysis followed by multiple analysis was used to identify the statistical significance between groups. *: *p* ≦ 0.05; **: *p* ≦ 0.01; ***: *p* ≦ 0.001; ****: *p* ≦ 0.0001; ns: no-significance.

**Figure 8 ijms-23-12100-f008:**
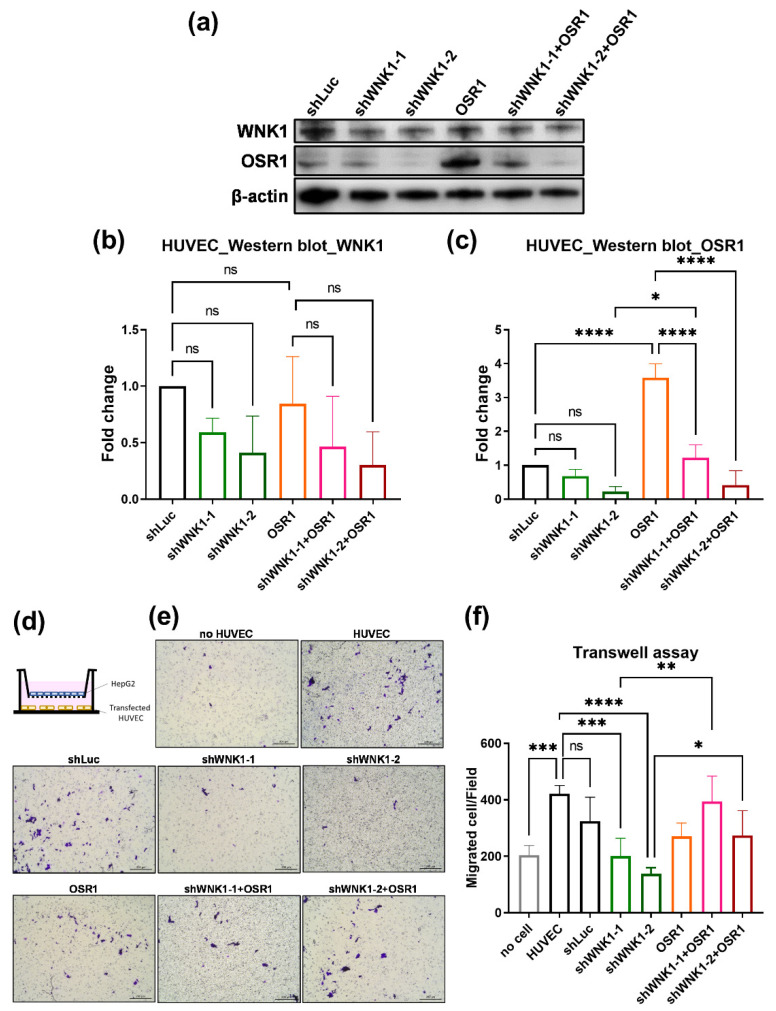
Knockdown of WNK1 in HUVEC cells significantly decreases the co-cultured hepatoma cell migration, and OSR1 overexpression can rescue the reduced cell migration caused by shWNK1 knockdown in endothelial cells. (**a**) The protein levels of WNK1 and OSR1 with WNK1 knockdown and/or OSR1 overexpression in HUVEC cells were detected by Western blot. (**b**) Quantification of WNK1 Western blot indicated that knockdown WNK1 (shWNK1-1 and shWNK1-2) in HUVEC cells significantly decreases the WNK1 protein levels. (**c**) Quantification of OSR1 Western blot indicated that overexpression of OSR1 in HUVEC cells significantly increases the OSR1 protein level, knockdown WNK1 (shWNK1-1 and shWNK1-2) in HUVEC cells significantly decreases the OSR1 protein levels. (**d**) HUVEC cells modulated the expression of WNK1 and OSR1 and were seeded in the lower chamber while HepG2 hepatoma cells were loaded in the upper chamber of the transwell plate. (**e**) Representative images of hepatoma cell migration. The scale bar is 200 µm in length. (**f**) Quantified data of the transwell migration assay shows that HUVEC cells with WNK1 knockdown significantly decrease the migrated hepatoma cells, indicating that WNK1 plays an important role in endothelial cells for stimulating hepatoma cell migration. Overexpression of OSR1 in shWNK1 knockdown HUVEC cells can revert the reduced hepatoma cell migration caused by shWNK1, indicating that OSR1 is downstream of endothelial WNK1 promoting hepatoma cell migration. We performed three replicates for each Western blot and transwell analysis. One-way ANOVA analysis followed by multiple analysis of variance was used to identify the statistical significance between groups. *: *p* ≦ 0.05; **: *p* ≦ 0.01; ***: *p* ≦ 0.001; ****: *p* ≦ 0.0001; ns: no-significance.

**Figure 9 ijms-23-12100-f009:**
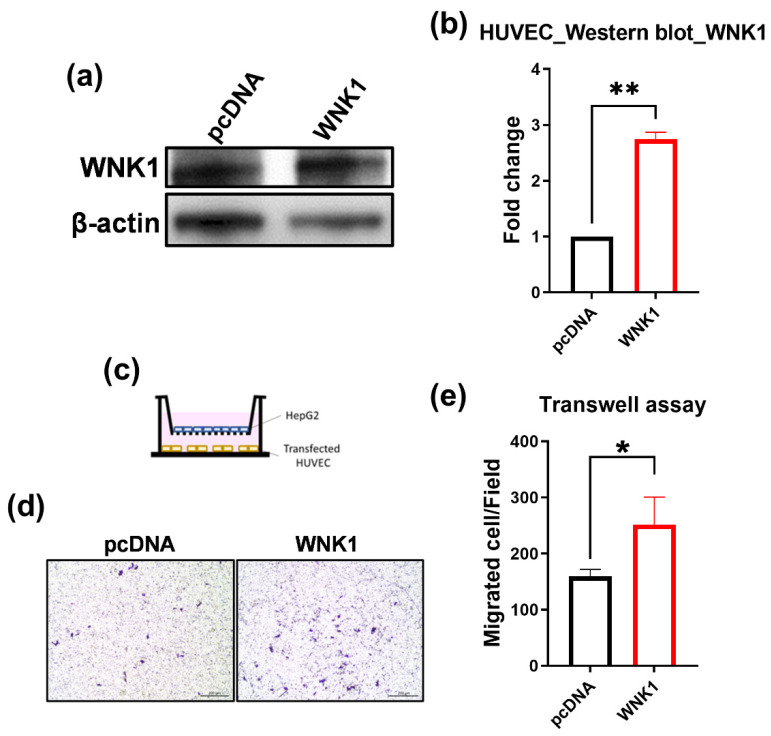
Overexpression of WNK1 in HUVEC cells significantly increases the migration of HepG2 cells. (**a**) The protein levels of WNK1 were examined by Western blot. (**b**) Quantification of Western blot indicates that overexpression of WNK1 in HUVEC cells significantly increases the WNK1 protein levels. (**c**) HUVEC cells with WNK1 overexpression were seeded in the lower chamber while HepG2 hepatoma cells were loaded in the upper chamber of the transwell plate. (**d**) Representative images of hepatoma cell migration. The scale bar is 200 µm in length. (**e**) Quantified data of the transwell migration assay shows that WNK1 overexpression in HUVEC cells significantly increases the migrated HepG2 cells, indicating that WNK1 plays an important role in endothelial cells for stimulating hepatoma cell migration. We performed three replicates for each Western blot and transwell analysis. Unpaired Student’s *t*-test (two tailed test) was used for statistical analysis. *: *p* ≦ 0.05; **: *p* ≦ 0.01.

**Figure 10 ijms-23-12100-f010:**
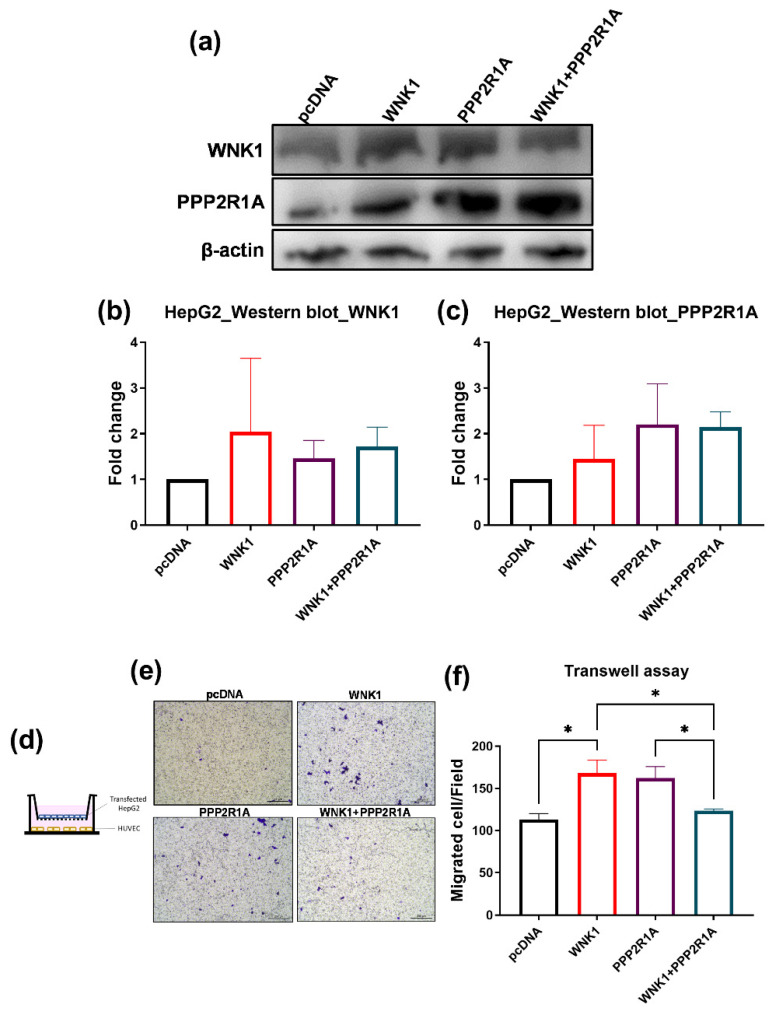
Overexpression of WNK1 in HepG2 cells significantly increases the migrated cells, and the co-expression of PPP2R1A can decrease the migrated cells caused by WNK1 overexpression in hepatoma cells. (**a**) The protein levels of WNK1 and PPP2R1A were examined by Western blot. (**b**) Western blot indicates that overexpression of WNK1 in HepG2 cells significantly increases the WNK1 protein levels. (**c**) Overexpression of PPP2R1A in HepG2 cells significantly increases the PPP2R1A protein levels. (**d**) The modulated expression of WNK1 and PPP2R1A in HepG2 hepatoma cells that were seeded in the upper chamber while HUVEC cells were loaded in the lower chamber of the transwell plate. (**e**) Representative images of hepatoma cell migration. The scale bar is 200 µm in length. (**f**) Quantified data of the transwell migration assay reveal that overexpression of WNK1 in HepG2 cells significantly increases the migrated cells, indicating that WNK1 plays an important role in hepatoma cells for stimulating hepatoma cell migration. Overexpression of PPP2R1A in HepG2 cells can revert the increased migrated cells caused by WNK1, indicating that PPP2R1A is a downstream effector for hepatoma cell migration. We performed three replicates for each Western blot and transwell analysis. One-way ANOVA analysis followed by multiple analysis of variance was used to identify the statistical significance between groups. *: *p* ≦ 0.05.

**Figure 11 ijms-23-12100-f011:**
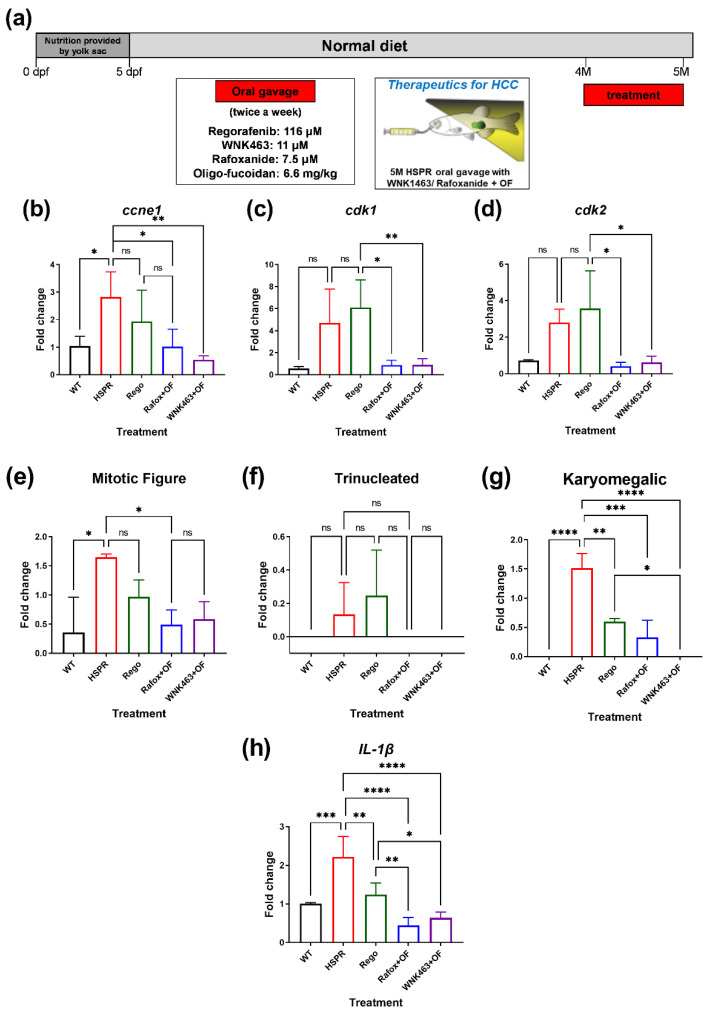
WNK1–OSR1 axis inhibitors combined with oligo-fucoidan attenuate HCC proliferation. (**a**) The treatment scheme of WNK1 and OSR1 inhibitors combined with oligo-fucoidan in [*HBx,src,p53^−/−^,RPIA*] HCC transgenic fish. The expression of proliferation markers (**b**) *ccne1*, (**c**) *cdk1*, and (**d**) *cdk2* in various combinations of drug-treated 5-month-old [*HBx,src,p53^−/−^,RPIA*] (HSPR) transgenic fish compared to WT. Rego: Regorafenib, Rafox: Rafoxanide as OSR1 inhibitor; WNK463 as WNK1 inhibitor; OF: oligo-fucoidan. The quantification of the histopathological features derives from H&E stain: (**e**) mitotic figures, (**f**) trinucleated hepatocytes, (**g**) karyomegalic cells in various combinations of drug-treated 5-month-old [*HBx,src,p53^−/−^,RPIA*] transgenic fish compared to WT. (**h**) The expression of cell senescence marker *il1β* in various combinations of drug-treated 5-month-old [*HBx,src,p53^−/−^,RPIA*] (HSPR) transgenic fish compared to WT. We performed three replicates, and for each qPCR, there are triplicates for each sample. Five adult fish are included in each experiment. One-way ANOVA analysis followed by multiple analysis of variance was used to identify the statistical significance between groups. *: *p* ≦ 0.05; **: *p* ≦ 0.01; ***: *p* ≦ 0.001; ****: *p* ≦ 0.0001; ns: no-significance.

**Figure 12 ijms-23-12100-f012:**
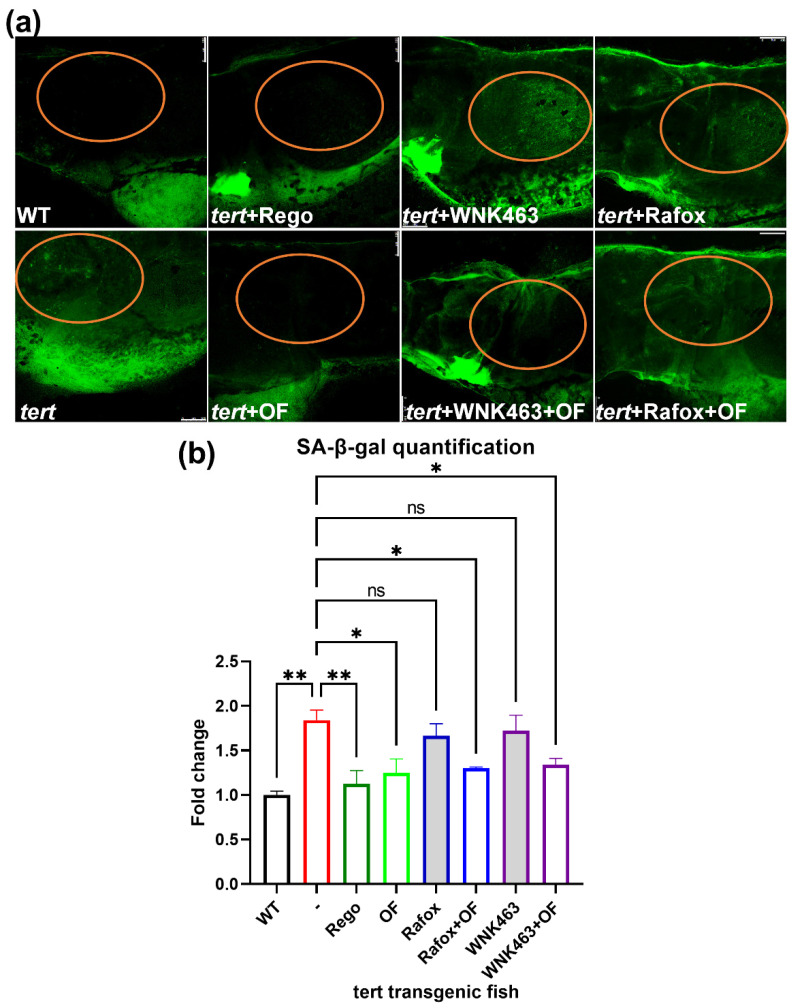
Oligo-fucoidan reduces the elevated senescence-associated β-galactosidase (SA-β-gal) expression by WNK1–OSR1 axis inhibitors in *tert* transgenic fish. We use Tg(fabp10a:tert; myl7:GFP) transgenic fish to further verify whether drugs can prevent cellular senescence. These fish were fed with a normal diet from 5 dpf to 10 dpf old and treated with several drugs: Regorafenib (0.5 μM), oligo-fucoidan (0.3 mg/mL), WNK463 (0.625 μM) with or without oligo-fucoidan, Rafoxanide(0.3 μM) with or without oligo-fucoidan, and were then fixed and stained on 11 dpf. (**a**) The SA-β-gal expression in various combinations of drug-treated tert transgenic fish after various drug treatments. WT: AB(WT) fish as control, *tert*: Tg(fabp10a:tert; myl7:GFP) fish; OF: oligo-fucoidan-treated tert transgenic fish; WNK463: WNK1 inhibitor WNK463-treated *tert* transgenic fish; WNK463+OF: WNK463 plus oligo-fucoidan-treated *tert* transgenic fish; Rafox: OSR1 inhibitor Rafoxanide-treated *tert* transgenic fish; Rafox+OF: Rafoxanide plus oligo-fucoidan-treated *tert* transgenic fish. The scale bar is 100 µm in length. The orange circle indicates the location of the liver. (**b**) Quantification of SA-β-gal staining. For each experiment, we performed three replicates. Ten larvae were included in each experiment. One-way ANOVA analysis followed by multiple analysis of variance was used to identify the statistical significance between groups. *: *p* ≦ 0.05; **: *p* ≦ 0.01; ns: no-significance.

**Table 1 ijms-23-12100-t001:** Transgenic fish lines used in this study.

Fish Lines	Promoter	Expressed Gene	Phenotype
AB wild-type	-	-	control
Tg(fli1:EGFP)	fli1	EGFP	Green fluorescence in the vessels.
Tg(fabp10a:HBx,src,RPIA;myl7:EGFP)xtp53^zdf1/zdf^	fabp10a (liver-specific promoter)	HBx, src, RPIA tp53^zdf1/zdf^	Green fluorescence in heart. Develops HCC at 5 months of age.
Tg(fli1:wnk1a;myl7:EGFP)	fli1	wnk1a	Green fluorescence in heart. Overexpressed wnk1a in the vessels.
Tg(fli1:EGFP)xTg(fli1:CreERT2;myl7:EGFP)xTg(loxP-wnk1a-DsRed-loxP)	fli1	wnk1a CreERT2 loxP-wnk1a-DsRed-loxP	Vessels-specific knockout the *wnk1a* by adding the RU486 activates CreERT2.
Tg(fli1:EGFP)xTg(fli1:wnk1a)	fli1	wnk1a, EGFP	
Tg(fabp10a:tert;myl7:EGFP)xtp53^zdf1/zdf^	fabp10a (liver-specific promoter)	tert tp53^zdf1/zdf^	Green fluorescence in heart. Develops HCC at 15, 30 days of age.
Tg(fabp10a:tert;myl7:EGFP)	fabp10a (liver-specific promoter)	tert	Green fluorescence in heart. Develops HCC at 15 days of age.

**Table 2 ijms-23-12100-t002:** Primer information.

Gene Name	Primer Name	Sequencing
*STK39*	h-Q-stk39_F	5′-CATGAGTCAGTGCAGCCATC-3′
	h-Q-stk39_R	5′-TGTGTTCTCCTCGGTTGACA-3′
*OXSR1*	h-Q-oxsr1_F	5′-AAAGACCTTTGTTGGCACCC-3′
	h-Q-oxsr1_R	5′-AGGATCGTTCTGCAGTGTCA-3′
*PPP2R1B*	h-Q-ppp2r1b_F	5′-GTCCTGACTTTGCCCACTGT-3′
	h-Q-ppp2r1b_R	5′-GAACCAATCCCCACTTGCTA-3′
*PPP2R1A*	h-Q-PPP2R1A_F	5′-TGACTGTCGGGAGAATGTGA-3′
	h-Q-PPP2R1A_R	5′-GGGAGAGAGACCCATGATGA-3′
*stk39*	z-Q-stk39_F	5′-TGGACACCTGCACAAAACTG-3′
	z-Q-stk39_R	5′-TCGTTTTCTTTGACCCTGCG-3′
*oxsr1a*	z-Q-oxsr1a_F	5′-AGGTGGCCATTAAACGCATC-3′
	z-Q-oxsr1a_R	5′-GCAACTTCATGACCAGCCAA-3′
*oxsr1b*	z-Q-oxsr1b_F	5′-CATCAAACGCATCAATCTGG-3′
	z-Q-oxsr1b_R	5′-CGGTCTTGTGTTCACCCTTT-3′
*ppp2r1ba*	z-Q-ppp2r1baF	5′-TGGCAACAGTTGAAGAGACG-3′
	z-Q-ppp2r1baR	5′-AGAGCCCACAAGCAGAGGTA-3′
*ppp2r1bb*	z-Q-ppp2r1bbF	5′-AGACTTGGAGGCTCTGGTCA-3′
	z-Q-ppp2r1bbR	5′-GGTCTCCCTGCTGTCTTCAG-3′
*ccne1*	z-ccne1_F	5′-CATGCCAAGCAAGAAAGTGCTA-3′
	z-ccne1_R	5′-GTGCTGGGAACACCTTCAGT-3′
*cdk1*	z-cdk1_F	5′-CTCTGGGGACCCCTAACAAT-3′
	z-cdk1_R	5′-CGGATGTGTCATTGCTTGTC-3′
*cdk2*	z-cdk2_F	5′-GGGCACTTTTGACATGGAGT-3′
	z-cdk2_R	5′-GTGCTGGGAACACCTTCAGT-3′
*actin*	z-actin_F	5′-CTCCATCATGAAGTGCGACGT-3′
	z-actin_R	5′-CAGACGGAGTATTTGCGCTCA-3′
*il1β*	z-Q-il1b_	5′-CGCTCCACATCTCGTACTCA-3′
	z-Q-il1b_R	5′-ATACGCGGTGCTGATAAACC-3′

**Table 3 ijms-23-12100-t003:** Components of reverse-transcription reaction.

Component	Volume per Reaction (μL)
5× iScript Reaction Mix	4
iScript Reverse Transcriptase	1
RNase-free water	variable
RNA template(1 μg)	variable
Total volume	20

**Table 4 ijms-23-12100-t004:** Components of QPCR reaction.

Component	Volume per Reaction (μL)
cDNA (dilute with RNase-free water)	3.8
Primer (2.5 μΜ of forward and reverse primer)	1.2
2X SYBR Green (Catalog #: 4385618, Thermo Scientific, Waltham, MA, USA)	5

**Table 5 ijms-23-12100-t005:** PCR program.

	Temperature	Time	Cycle
Hold stage	95 °C	3 min	1
PCR stage	95 °C	1 s	40
	60 °C	20 s	
Melt curve stage	95 °C	15 s	1
	60 °C	1 min	
	95 °C	15 s	

## Data Availability

Not applicable.

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
