# Peer review of "WNK1–OSR1 Signaling Regulates Angiogenesis-Mediated Metastasis towards Developing a Combinatorial Anti-Cancer Strategy"

_ijms, 2022, doi:10.3390/ijms232012100_

Round 1

Reviewer 1 Report

This is very extensive and well-executed study investigating the role of WNK1 pathway in hepatoma cells and tumor angiogenesis. Please find below comments to help improve the manuscript.

Introduction:

1.       Line 49: Cardiovascular disease is the number one cause of death globally. Please adjust your statement to “Cancer remains one of the top leading causes of death globally”, or adjust to include actually/estimated numbers of deaths worldwide.

2.       Line 52-53: the fact that liver cancer tends to be a highly vascularized tumor, is not an indication per se that targeting tumor angiogenesis would be a good anti-cancer strategy. Instead, one could state: “as such, targeting tumor-induced angiogenesis has been suggested as a potential anti-cancer strategy.

3.       Line 54-55: Change sentence to: “In comparison to cancer cells that exhibit heterogeneity and genomic instability, DNA of endothelial cells is…”

4.       Please have introduction carefully read by somebody fluent in English to assure correct grammar.

5.       Line 59: Include correct abbreviation after “with-no-lysine kinase” (WNK)

Results:

1.       Line 240-241: the authors state that wnk1a is upregulated from 1dpi to 3 dpi, while fig 3a shows only upregulation for 1 dpi and 2 dpi. Please correct this. Why do you think it is not up at 3 dpi?

2.       Line 286-287: Change sentence to “…was found to act as an oncogene in many cancers.”

3.       Figure 5: I believe this data is presented in the wrong manner. Figure 5a should be the expression of ccne in tert fish at 15dpf and 30dpf. The WT fish for Tert are possibly different than the wt fish for tertxp53 and thus you cannot compare fold change between 2 different strains. Fig 5b and fig 5c would subsequently be expression of cdk1 and cdk2 for tert fish at 15 and 30 dpf. Fig 5d-f would be the same data but for tert x p53. WT should be age and strain matched.

4.       How was data collected and analyzed for Fig 5g-j?

5.       Figure 6: what constitutes WT? Is this age-strain matched?

6.       Figure 7: what defines the error bars?

7.       Fig 8d should be Fig 8a, as the success of the knockdown determines how the other graphs in this figures are interpretated.

8.       Fig 9e: Is this fold change normalized to beta-actin? Fig 9d-e should be shown at the beginning of the figure as the interpretation of the remainder of the figure relies on this data.

9.       Line 448-449: please explain this observation/conclusion? PPP2R1A alone also increases cell migration of HepG2 cells, so why would PPP2R1A in combination with WNK1 no do the same?

1

Discussion:

1.       Line 592: PPP2R1A can rescue cell migration induced by overexpressed WNK1. Not the case with WT WNK1 levels based on Figure 10c.

2.       Why is stk39 change in zebrafish different than in HUVEC (Fig 3 vs Fig 4)?

Methods:

1.       Section 4.1: Please provide more detail about the different strains used. Perhaps it would be useful to put this information in a table format so that it is easier to comprehend what the phenotype is of each of these strains.

2.       Section 4.3 Cell culture: where did cell lines come from? Were HUVEC immortalized cells or primary cells? If primary, what passages were used?

3.       Section 4.4 Xenotransplantation: Could you please explain why embryos were used for the xenograft model rather than adult zebrafish? My concern would be that embryos, who are still in active development/growth stage, create a very different environment for tumor cells than adult fish, and thus might introduce factors that are irrelevant for an aging condition.

4.       Section 4.10 Western Blot analysis: Western blot is not used to determine transfection efficiency. Change the start of the paragraph to: “To determine effective knockdown of WNK1 through transfection and the impact on downstream mediators, we performed western blot analysis for XXX.”

5.       Section 4.12: what are the larva? Are these the zebrafish embryos? Also, larva and liver tissue should be in plural.

6.       4.13 Statistical analysis: it is stated that Student t-test is used for statistical analysis. In all the figures, except for Figure 9) three or more groups are being compared. Student T-test is not an appropriate test for this analysis. Instead, you should be using a one-way ANOVA. If significant, the one-way ANOVA should be followed by a posthoc test to identify where the differences lay. Was 2-sided test used?

7.       Please provide information on how many replicates each experiment was performed and how often experiments were repeated, or how many zebrafish were included in each experiment? In other words, please explain what constitutes the error bars in all the experiments.

Reviewer 2 Report

Introduction seems a bit too long that could be shortened.

In view of marked endothelial cell heterogeneity and arterial versus venous endothelial cell differences it is debatable how representative is use of HUVEC cells in such studies.

There appears to be a lot of qPCR data but very little immunocytochemical analysis that could have improved this study.

Round 2

Reviewer 1 Report

The authors have made significant improvements to the manuscript. The only concern that is left is that the wrong statistical test is carried out throughout the manuscript. A student's T-test is not appropriate when comparing 3 or more groups. Instead, a One-way ANOVA needs to be carried out. If ANOVA shows statistical significance, a post-hoc test will identify what group is different from the others. 

Author Response

Thank you for your constructive comments, we have used One-way ANOVA for all the figures except Figure 9 which only have two groups. If ANOVA shows statistical significance, we used a post-hoc test to identify what group is different from the others. Please see attached revised manuscript with all the updated figures.